# Visualization of cytosolic ribosomes on the surface of mitochondria by electron cryo-tomography

Vicki AM Gold[1,2,3,*,†] iD, Piotr Chroscicki[4,†], Piotr Bragoszewski[4] & Agnieszka Chacinska[4,5,**] iD

## Abstract

We employed electron cryo-tomography to visualize cytosolic ribosomes on the surface of mitochondria. Translation-arrested ribosomes reveal the clustered organization of the TOM complex, corroborating earlier reports of localized translation. Ribosomes are shown to interact specifically with the TOM complex, and nascent chain binding is crucial for ribosome recruitment and stabilization. Ribosomes are bound to the membrane in discrete clusters, often in the vicinity of the crista junctions. This interaction highlights how protein synthesis may be coupled with transport. Our work provides unique insights into the spatial organization of cytosolic ribosomes on mitochondria.

**Keywords** electron cryo-tomography; mitochondrial protein import; ribosomes; TOM complex; translation

**Subject Categories** Membrane & Intracellular Transport; Protein Biosynthesis & Quality Control; Structural Biology

## Introduction

Historically, cytosolic ribosomes were thought to exist in two main pools, a free solution state and a endoplasmic reticulum (ER) membrane-bound state [1,2], both recently visualized *in situ* [3]. The membrane-bound ribosomes are engaged in a well-orchestrated process, in which protein synthesis is mechanistically coupled with protein translocation into the ER. This so-called co-translational mode of transport utilizes mechanisms that lead to translational stalling and precise positioning of the ribosomes at the ER membrane translocon, the Sec complex [4–8].

Mitochondria constitute an important bioenergetic, metabolic, and signaling hub, and the biogenesis of mitochondrial proteins is an important process that determines the organelle's function. The mitochondrial proteome of the yeast *Saccharomyces cerevisiae* is composed of ~900 proteins [9,10]. Almost all of them (99%) are nuclear encoded, despite the presence of mitochondrial DNA. Mitochondria-destined precursor proteins are synthesized on cytosolic ribosomes and are actively imported through the translocase of the outer membrane (TOM) complex. TOM forms a common entry gate for mitochondrial precursor proteins that are subsequently targeted to various mitochondrial locations [11–17]. Mitochondria have a double-membrane structure; thus, proteins destined for the mitochondrial matrix or inner membrane are subsequently transported through one of two protein translocases of the inner membrane. Proteins that possess positively charged N-terminal presequences are substrates for the translocase of the inner membrane (TIM23) complex [11–17].

For decades, it has been known that precursor proteins can be imported into mitochondria post-translationally, after complete synthesis in the cytosol or *in vitro* in a ribosome-free system [13,18–20]. Meanwhile, cytosolic ribosomes were detected in the vicinity of mitochondria by electron microscopy (EM), suggesting a role for co-translational import [21,22]. Additionally, various independent approaches have shown an enrichment of mRNAs encoding mitochondrial proteins, either on the mitochondrial surface or in close proximity, both in yeast [23–28] and human cells [29,30]. The process of mRNA targeting to mitochondria is not well characterized, but COP1 and the outer membrane-associated protein Puf3, which binds 3′ non-coding sequences of mRNAs, were identified to play a role in this process [31–33]. Mitochondrial surface-localized mRNA molecules were also found to be active as templates for protein synthesis [28]. The mechanism of ribosome-nascent chain complex (RNC) recruitment to mitochondria was also investigated by a *de novo* ribosome binding assay [34–38]. In summary, there is a great deal of data in support of localized synthesis of proteins at the mitochondrial outer membrane, yet the co-localization of cytosolic ribosomes with TOM complex has never been shown to date.

Electron cryo-tomography (cryoET) is a technique by which proteins or complexes may be studied *in situ*. Samples are preserved by cryo-fixation, imaged in the electron microscope, and structures can be determined by subtomogram averaging (StA) [39]. The

1  Department of Structural Biology, Max Planck Institute of Biophysics, Frankfurt am Main, Germany
2  Living Systems Institute, University of Exeter, Exeter, UK
3  College of Life and Environmental Sciences, Geoffrey Pope, University of Exeter, Exeter, UK
4  The International Institute of Molecular and Cell Biology, Warsaw, Poland
5  Centre of New Technologies, University of Warsaw, Warsaw, Poland
   *Corresponding author. Tel: +44 1392 727454; E-mail: v.a.m.gold@exeter.ac.uk
   **Corresponding author. Tel: +48 22 55 43600; E-mail: a.chacinska@cent.uw.edu.pl
   †These authors contributed equally to this work

post-translational route for protein import into mitochondria was previously studied by this method, revealing details of TOM-TIM23 supercomplex localization and distribution [40]. In this work, we isolated native mitochondria with bound ribosomes, confirming earlier reports of localized translation on the mitochondrial surface. Ribosome numbers were low; thus, we devised a method to isolate sufficiently high numbers of mitochondria with associated ribosomes (MAR) by translational arrest with cycloheximide (CHX) treatment. Samples were characterized biochemically and imaged using cryoET and StA. This demonstrated that a specific interaction between ribosomes and the TOM complex occurs and that nascent chain binding is crucial for ribosome recruitment and stabilization on the mitochondrial outer membrane. Ribosomes, which mark the position of TOM complexes, were visualized on the mitochondrial surface in discrete clusters, often within the vicinity of the crista junctions (CJs), providing a long awaited view of cytosolic ribosomes bound to mitochondria.

# Results

## Cytosolic ribosomes co-purify with mitochondria and can be stabilized on the outer membrane

In standard preparations of isolated yeast mitochondria, cytosolic ribosomes are not observed bound to the outer membrane by cryoET (Fig 1A and D) [40]. We first investigated whether mitochondria-bound RNCs could be enriched with magnesium acetate $(Mg(OAc)_2)$, as $Mg^{2+}$ ions are essential for ribosome and RNC stabilization. In mitochondrial preparations isolated in the presence of

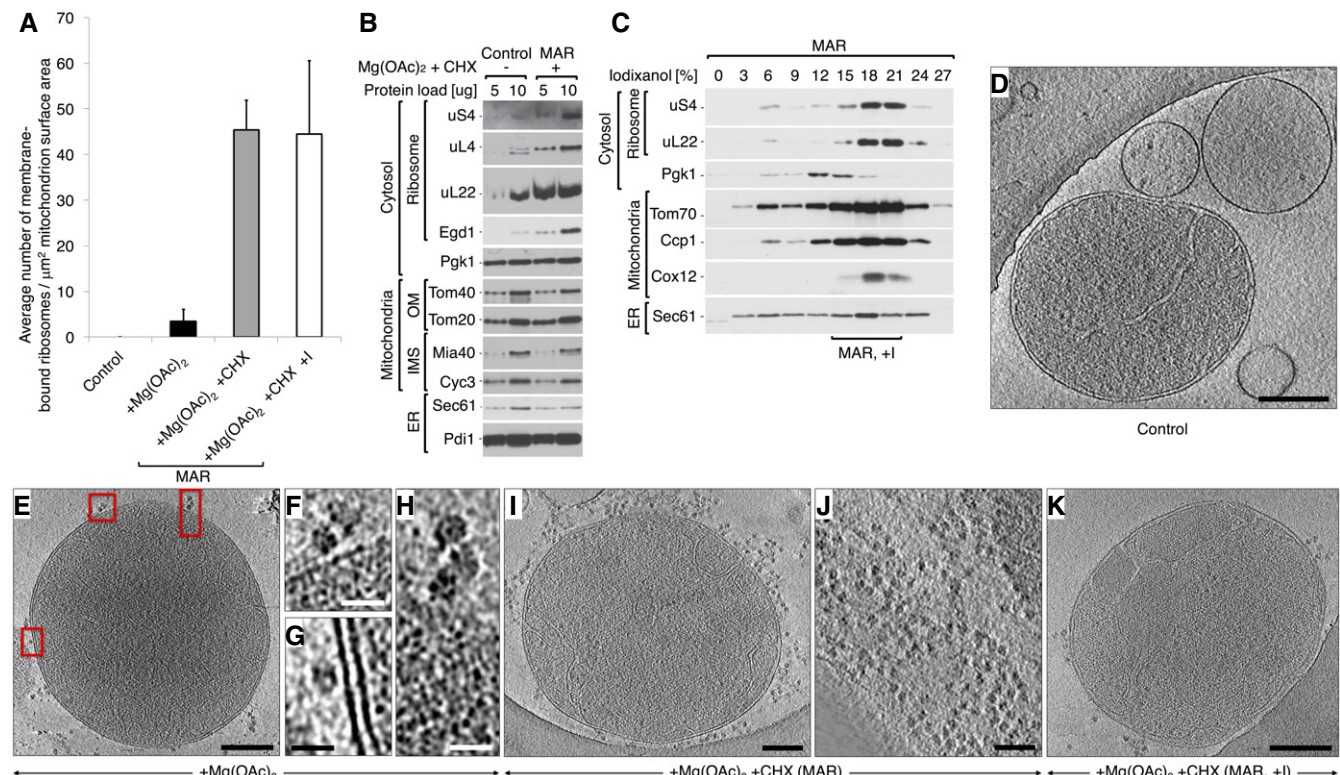

**Figure 1. Mitochondria are enriched with ribosomes after CHX treatment.**

A    Average number of ribosomes bound to mitochondria for control ($-Mg(OAc)_2$ $-CHX$), $+Mg(OAc)_2$ only, and two $+Mg(OAc)_2$ $+CHX$ (MAR) samples, from a crude isolation and iodixanol purification (+I). Data are plotted as the mean number of ribosomes $\pm$ SEM. $n = 28$ mitochondria, combined from > 10 independent sample preparations.

B    The steady-state protein levels of isolated crude mitochondria are shown for control ($-Mg(OAc)_2$ $-CHX$) and MAR ($+Mg(OAc)_2$ $+CHX$) samples. Ribosomal proteins co-isolate with mitochondria under ribosome-stabilizing conditions ($+Mg(OAc)_2$ $+CHX$). IMS, intermembrane space; OM, outer membrane.

C    Fractionation of MAR samples in a 0–27% iodixanol step gradient. Iodixanol gradient-purified MAR (MAR, +I) were isolated from 15 to 21% iodixanol layers. Co-sedimentation of a group of 80S ribosomes with mitochondria indicates their stable interaction.

D–K    Corresponding example tomographic slices for the data shown in (A). (D) Control ($-Mg(OAc)_2$ $-CHX$) mitochondria are not associated with ribosomes. Scale bar, 0.2 μm. (E) Samples treated with $+Mg(OAc)_2$ only show ribosomes (boxed) bound to mitochondria in a few cases. Scale bar, 0.3 μm. (F–H) Enlargement of the boxes shown in (E). Scale bars, 20 nm. (I) Crude preparation of a MAR ($+Mg(OAc)_2$ $+CHX$) sample shows many ribosomes bound to mitochondria, but also in (J), a high background of free cytosolic ribosomes that distort accurate analysis. Scale bars, 0.2 μm and 0.1 μm, respectively. (K) Analysis of the iodixanol gradient-purified MAR sample shows that ribosomes remain stably bound to mitochondria after centrifugation. The background level of free ribosomes and ER membranes is reduced. Scale bar, 0.2 μm.

Data information: In (B, C), samples were analyzed by SDS–PAGE followed by immunodecoration with specific antisera. ER, endoplasmic reticulum.

$Mg(OAc)_2$, we were able to clearly identify ~3 ribosomes (per $\mu m^2$ mitochondrion surface area) bound to mitochondria on average (Figs 1A, E–H and EV1A), confirming the stabilizing effect of $Mg^{2+}$ and the potential for co-translational protein import to mitochondria. The number of bound ribosomes was, however, too low for comprehensive structural or statistical analysis. CHX is known to block the translocation step of translational elongation, thus stabilizing RNCs [41–43]. CHX treatment also alters the kinetics of protein translation and targeting to mitochondria, but nevertheless could be used as a means to visualize ribosomes arrested on the outer membrane. Treatment of the cells and CHX inclusion in the buffers for mitochondrial isolation had the effect of increasing the number of ribosomes to ~45 (per $\mu m^2$ mitochondrion surface area) on the mitochondrial membrane, a 15-fold increase (Figs 1A and I, and EV1A). The $Mg^{2+}$ and CHX-treated mitochondria are subsequently referred to as MAR. The steady-state protein levels of isolated control and MAR samples were subsequently analyzed and confirmed observations made by cryoET (Fig 1B). Accordingly, protein markers of both the 60S (uL22 and uL4) and the 40S ribosome (uS4) were significantly increased in the MAR sample compared to the control. Additionally, the level of Egd1, a β subunit of the nascent polypeptide-associated complex (NAC), was also increased. NAC binds to the ribosome and acts to protect the nascent chain and facilitate folding [44–47]. Marker proteins for mitochondria (Tom40, Tom20, Mia40, Cyc3), cytosol (Pgk1), and ER (Sec61, Pdi1) remained in equal amounts between control and MAR samples (Fig 1B).

Mitochondria exist in a dynamic network and interact closely with other organelles in the cell, most notably the ER [48]. Thus, mitochondria isolated by differential centrifugation inevitably co-purify with ER membranes of similar density. Consequently, CHX treatment also had the effect of increasing the overall level of ribosomes, which were observed either bound to ER membranes, or were free in solution (Fig 1J). Due to the heterogeneous nature of different populations of ribosomes in tomograms, StA of the mitochondria-bound population was extremely challenging. Therefore, an iodixanol gradient purification step [40] was included to remove soluble material and a proportion of rough ER membranes, as visualized by Western blot analysis (Fig 1C). Mitochondrial marker proteins (Tom70, Ccp1, Cox12) were mostly enriched in the same fractions as the ribosomal marker proteins uS4 and uL22 (15–21%), which were pooled for further analysis by cryoET. The purification step removed a portion of free cytosolic ribosomes and rough ER membranes (Figs 1C and K, and EV1B) but importantly did not affect the number of ribosomes bound stably to the outer membranes of mitochondria (Figs 1A and EV1A).

## Ribosome binding to mitochondria is dependent on protein import and involves the TOM complex

The TOM complex is the exclusive entry gate for imported mitochondrial proteins. Therefore, to test the specificity of ribosomes binding to mitochondria, we assessed the cytosolic ribosome interaction with the TOM complex in MAR samples. Affinity purification of the TOM complex, via its Histidine10-tagged core protein Tom22, demonstrated that the ribosomal protein marker uL22 and the ribosome-localized Hsp70 family chaperone Ssb1 could be co-purified (Fig 2A, lane 7). Pretreatment with EDTA resulted in a loss of the

ribosome–Tom22$_{HIS}$ interaction (Fig 2A, lane 8), due to the depletion of $Mg^{2+}$ ions and thus ribosome dissociation. To confirm this result, the ribosome–TOM complex interaction was further investigated by affinity purification of HA-tagged Tom40, the TOM complex component that forms the central pore of the translocase. This again demonstrated the co-purification of ribosomal protein uL22 from MAR samples (Fig EV2A, lane 4). Similarly, uL22 and Ssb1 were eluted with Tom40$_{HA}$ when high molecular weight (HMW) membranes, which are enriched membranes from mitochondria (Fig EV2B), were subjected to affinity purification (Fig 2B, lane 4).

The observed interaction of ribosomes with the TOM complex could feasibly be mediated by nascent chains of mitochondrial precursor proteins. To test this hypothesis, we analyzed ribosome association with mitochondria after dissipation of the electrochemical potential ($-\Delta\Psi$) of the mitochondrial inner membrane with carbonyl cyanide *m*-chlorophenyl hydrazone (CCCP). Precursors with N-terminal presequences and hydrophobic inner membrane proteins are known to require the $-\Delta\Psi$ for their import [13,14,17]. We observed a time-dependent reduction in the amount of ribosomes associated with HMW membranes in the samples treated with CCCP, as indicated by ribosome marker proteins (uS4, uL22, and Egd1) (Fig 2C and D). This change is likely a direct effect due to dissipation of the electrochemical potential of the mitochondrial inner membrane, but the secondary effects of CCCP cannot be fully excluded. It was shown previously that mitochondrial precursor proteins accumulate in the cytosol upon dissipation of the $-\Delta\Psi$ [19,49]. However, simultaneous treatment with CCCP and CHX did not reduce the amount of ribosomes in isolated MAR and HMW membrane samples (Fig EV2C and D, lane 4). This may indicate that CCCP does not affect the localization of ribosomes that are already stably bound to mitochondria.

We reasoned that ribosome-bound nascent chains are involved in RNC binding to the mitochondrial outer membrane. Thus, nascent chain release should cause ribosome dissociation from mitochondria. Puromycin is a commonly used translation inhibitor that competes with aminoacetylated tRNA at the ribosomal A site, causing premature translation termination and polypeptide release [50]. However, CHX inhibits mRNA translocation; thus, the use of CHX during MAR isolation would block nascent chain puromycilation [43,51]. For this reason, we applied hydroxylamine ($NH_2OH$) as an alternative nascent chain releasing agent [52,53], which is a small compound that can reach the ribosomal active site and break the peptidyl–tRNA bond. To confirm nascent chain release, we took advantage of RNCs harboring the nascent chain for Tim9 (directed to the intermembrane space), which was lacking a stop codon [54,55]. A radiolabeled nascent chain bound to ribosomes can be detected in a complex with tRNAs when analyzed by SDS–PAGE [54] (Fig EV2E, lane 1). As expected, incubation of RNCs containing Tim9 with hydroxylamine caused the aminolysis of tRNA-Tim9 complexes and formation of mature-size Tim9 protein (Fig EV2E, lane 2). Subsequently, hydroxylamine treatment of MAR samples resulted in ribosome dissociation from mitochondria upon treatment (Figs 2E lanes 2–5 and F, and EV2F lanes 2–5 and G). As expected, puromycin was not effective due to prior use of CHX during MAR isolation (Fig 2E, lane 6). To confirm nascent chain release by cryoET, we further subjected MAR samples preincubated with 1,5M hydroxylamine to centrifugation in an iodixanol gradient as before.

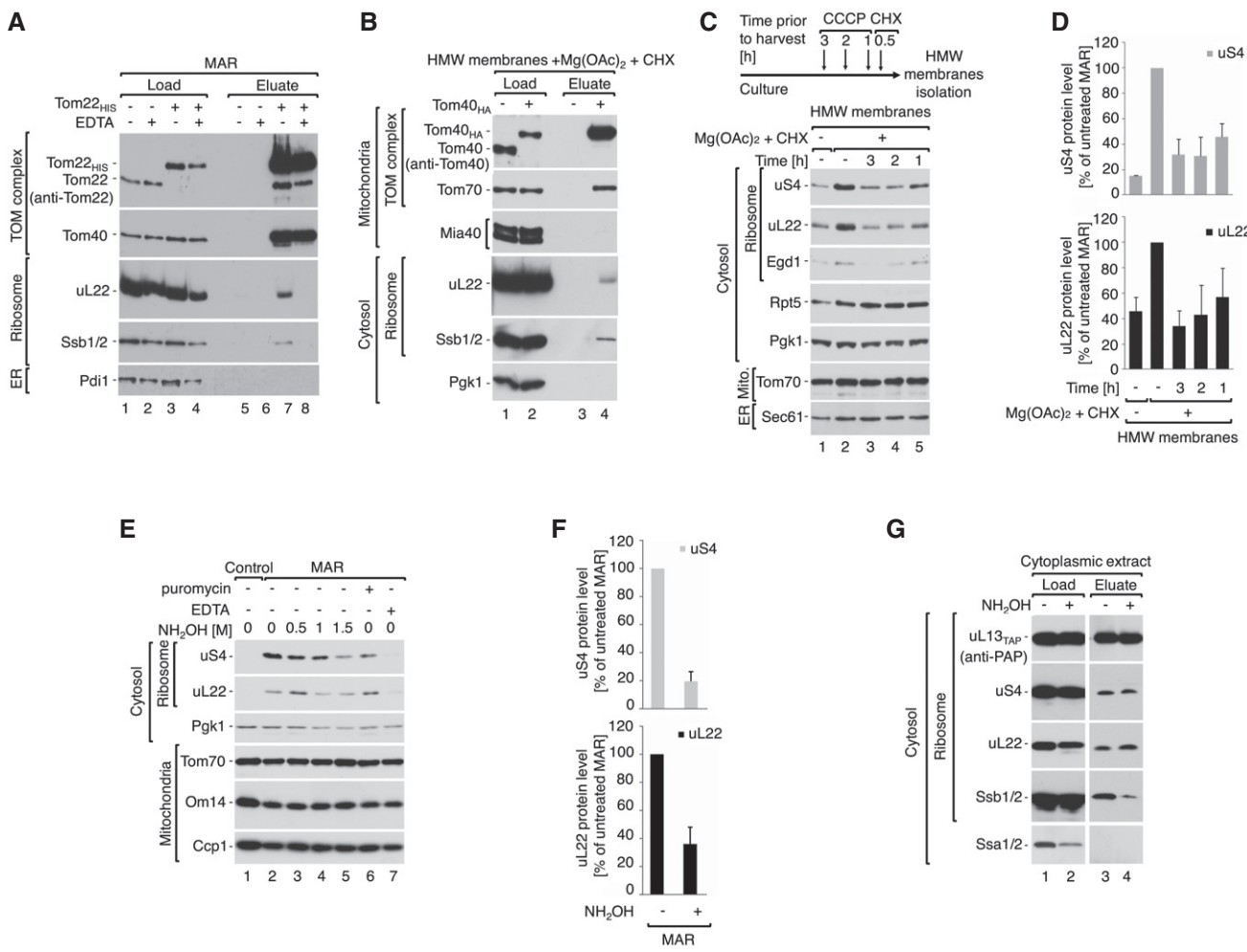

**Figure 2. Cytosolic ribosomes interact with the mitochondrial TOM translocase via the nascent chain.**

A, B   Cytosolic ribosomes co-purify with the TOM complex. (A) Immuno-affinity purification of Tom22$_{HIS}$ from digitonin-solubilized MAR. MAR were pre-treated with 25 mM EDTA and washed before solubilization. Load 2%; eluate 100%. (B) Immuno-affinity purification of Tom40$_{HA}$ from digitonin-solubilized HMW membranes. Load 1%; eluate 100%.

C, D   Dissipation of the electrochemical inner membrane potential inhibits ribosome recruitment to the mitochondrial surface. (C) The steady-state protein levels of HMW membranes isolated from cells that were either untreated, or treated with 10 μM CCCP for 3, 2 or 1 h prior to harvesting. Translation was inhibited with 50 μg/ml CHX for 30 min prior to harvesting. For analysis of protein levels after shorter CCCP treatment times see Fig EV2C and D. (D) Quantification of the ribosomal protein levels from samples shown in (C). The protein levels of uS4 and uL22 in MAR were set to 100%. Data are presented as the mean ± SEM. *n* = 3 biological replicates.

E–G   Ribosomes dissociate from mitochondria upon nascent chain release. (E) Protein levels in MAR samples upon treatment with nascent chain releasing agents: hydroxylamine and 3 mM puromycin. 25 mM EDTA was used as reference for ribosome clearance from MAR samples. (F) Quantification of the ribosomal protein levels from untreated MAR and after treatment with 1.5 M hydroxylamine shown in (E, lanes 2 and 5). The protein levels of uS4 and uL22 in untreated MAR were set to 100%. Data are presented as the mean ± SEM. *n* = 3 biological replicates. (G) TAP-tag affinity purification of ribosomes from the cytoplasmic fraction after hydroxylamine treatment from the uL13$_{TAP}$ strain. Hydroxylamine causes nascent chain release together with chaperone Ssb1/2, without affecting 80S ribosome structure. Load 4%; elution 100%.

Data information: In (A–C, E, G), samples were analyzed by SDS–PAGE and Western blotting using specific antisera.
Source data are available online for this figure.

The majority of ribosomal proteins (uS4, uL22) and ribosome-associated protein (Ssb1/2) were now detected in lighter fractions, similar to the cytosolic protein Pgk1 (Fig EV2H). Only 15 mitochondria-bound ribosomes (per μm² mitochondrion surface area) could now be identified in hydroxylamine-treated MAR samples, showing a 66% reduction compared to the untreated state (Fig EV2I). To exclude a negative effect of high hydroxylamine concentration on the ribosome 80S structure, we purified cytoplasmic ribosomes

preincubated with 1.5 M hydroxylamine using the TAP-tagged large ribosomal subunit uL13$_{TAP}$. The ribosomal proteins (uS4 and uL22) were detected in the eluate at the same level in the control as well as the hydroxylamine-treated sample (Fig 2G, lanes 3 and 4), confirming that hydroxylamine does not cause 80S ribosome disassembly. Interestingly, hydroxylamine caused dissociation of Ssb1/2 from the ribosome (Fig 2G, lane 4). In line with our findings, it was also shown that puromycin reduces the amount of ribosome-bound

Ssb1/2 due to nascent chain release [56]. To summarize, ribosomes bind to the TOM complex via the nascent chain in a $-\Delta\Psi$-dependent manner, and are sensitive to hydroxylamine, which specifically removes nascent chains from the ribosome.

## Mitochondria-bound ribosomes are specifically oriented for protein import

To investigate the 3D localization of ribosomes bound to mitochondria, iodixanol-purified MAR samples were investigated in detail by cryoET and StA (Figs 3 and 4). Two different populations of ribosomes could be clearly observed: The first was a distinct group located at the mitochondrial membrane (MAR-M, orange arrowheads in Fig 3A–C), and the second group was more peripherally associated (MAR-P, blue arrowheads in Fig 3A and C). In order to visualize ribosome distribution and their specific orientation with respect to the mitochondrial outer membrane, the MAR-M (1,215 subvolumes) and MAR-P (419 subvolumes) structures were determined by StA (Figs 3D and E, and EV3). Placing the MAR-M and MAR-P structures back into the 3D tomographic volume revealed a number of interesting details (Movie EV1). Firstly, both groups form discrete clusters on mitochondria (Fig 3F), in agreement with previous data reporting on the distribution of proteins arrested through TOM-TIM23 supercomplexes [40]. Ribosomes were also observed to group locally around a tubular section of one mitochondrion, which is possibly a fission constriction (Fig 3G–J) [57]. Interestingly, TOM-TIM23 arrested preproteins were previously found to cluster around a fusion septum [40], providing additional support for the idea that protein import sites occur at specific microdomains.

In the MAR-M population, ribosomes were clearly specifically oriented with the polypeptide exit tunnel pointing toward the outer membrane for import, often within the vicinity of the CJs (Fig 4A). Soluble MAR-P clusters were observed to associate with neighboring MAR-M groups and did not appear to adopt any specific orientations with respect to the membrane (Figs 3F and J, and 4B). In general, polysomes form clusters that translate mRNA simultaneously and form highly flexible structures [58–60]. On this basis, we suggest that ribosomes in the MAR-P group are polysomes, attached to MAR-M ribosomes through mRNA molecules (Fig 4B). In support of this, the amount of ribosomal proteins were reduced by ~50% in gradient-purified MAR treated with ribonuclease A (Fig 4C and D), consistent with our tomographic visualization of MAR-M and MAR-P populations on the ribosomal surface (compare Figs 3F with 4E and Fig 3I with J). Interestingly, Ssb1/2 was less sensitive to ribonuclease A treatment (32% reduction), showing its higher affinity to MAR-M ribosomes (Fig 4C and D). This is in line with our expectations, as nascent chains may be too short to emerge from the exit tunnel in some MAR-P to interact with Ssb1/2 [45,46,56,61].

## Using ribosomes to investigate clustering of the TOM complex

Based on the results from our biochemical data (Figs 2 and EV2), the ribosome could be used as a tag to mark the position of the TOM complex *in situ*. To investigate observed clustering of protein import sites on the mitochondrial surface in more detail, distance calculations were made between individual ribosomes and their

closest-neighbor using an established protocol [40]. This revealed that ~90% of TOM complexes exist in discrete clusters, marked by two or more ribosomes located < 50 nm apart (Fig 5A). For statistical analysis of ribosome numbers, the absolute values of both MAR-M and MAR-P populations on individual mitochondria were correlated with the surface area of the outer membrane. This revealed a linear correlation for both populations, with an average value of 157 MAR-M (TOM complexes) and 84 MAR-P per $\mu m^2$ outer membrane surface, respectively (Fig 5B). This is in line with the ~50% reduction in ribosomes previously observed after ribonuclease A treatment (Fig 4D), due to the loss of the MAR-P population. Many recent reports detail the relationship between the import machinery and the CJ [54,62–65]. To directly visualize the spatial relationship between the TOM complex and the CJ *in situ*, the distance between each MAR-M ribosome and its nearest CJ was calculated (Fig EV4). This was compared to previous data [40] (now visualized differently) showing the distribution of saturated TOM-TIM23 supercomplexes (Fig 5C). This analysis revealed that while both TOM and TOM-TIM23 supercomplexes tend to cluster preferentially around CJs, the TOM complex distribution is significantly broader than that of TOM-TIM23 (Fig 5D). Additional statistical analyses were performed to investigate the distribution of cluster sizes. For both data sets, < 15% of ribosomes existed as a single entity, and the major group size was between two and five ribosomes per cluster (Fig 5E and F). In the MAR-M population, ~5% of ribosomes existed in "superclusters", defined as a group of > 26 ribosomes. MAR-P clusters existed in groups of maximum 25 ribosomes, similar to that reported previously for cytosolic ribosomes observed in whole cells [59]; in this case, "superclusters" were not seen (Fig 5F).

## Comparison to ribosome tethering to the ER

From the same samples that were used for cryoET of MAR-M and MAR-P, 230 ER-bound ribosomes (ER-R) could also be identified for StA from the same tomograms (Figs 6A and B, and EV3). Visualization of the resulting average in a corresponding 3D tomographic volume also revealed discrete clusters on small vesicles (Fig 6C). However, as we only report on a small part of the ER-R population, detailed statistical analysis of clustering was not carried out. A small density could be observed to make a connection between ribosomes and the ER membrane (Fig 6D). By docking X-ray structures of yeast ribosomes [66] into the ER-R and MAR-M StA maps, the density was identified as rRNA expansion segment eS7$^L$a (Fig 6E). This is in agreement with previous reports of ER membrane-associated canine ribosomes [60]. Contra to the ER-R population, at this resolution eS7$^L$a is not seen to connect to the mitochondrial membrane (Fig 6F). No density was observed for rRNA expansion segment eS27$^L$ in either structure (Fig 6E and F), in line with previous reports of its extremely dynamic behavior [66].

The lack of protein or rRNA density between the ribosome and the mitochondrial membrane suggests that CHX-stabilized ribosomes could be tethered to the TOM complex by the nascent chain only. Analysis of the distances between MAR-M or ER-R populations and their corresponding membranes demonstrated the variability in tethering between the two groups. The average distance (measured from the base of the cleft between the 60S and 40S subunits to the membrane) was similar, at ~13 nm and ~12 nm, respectively

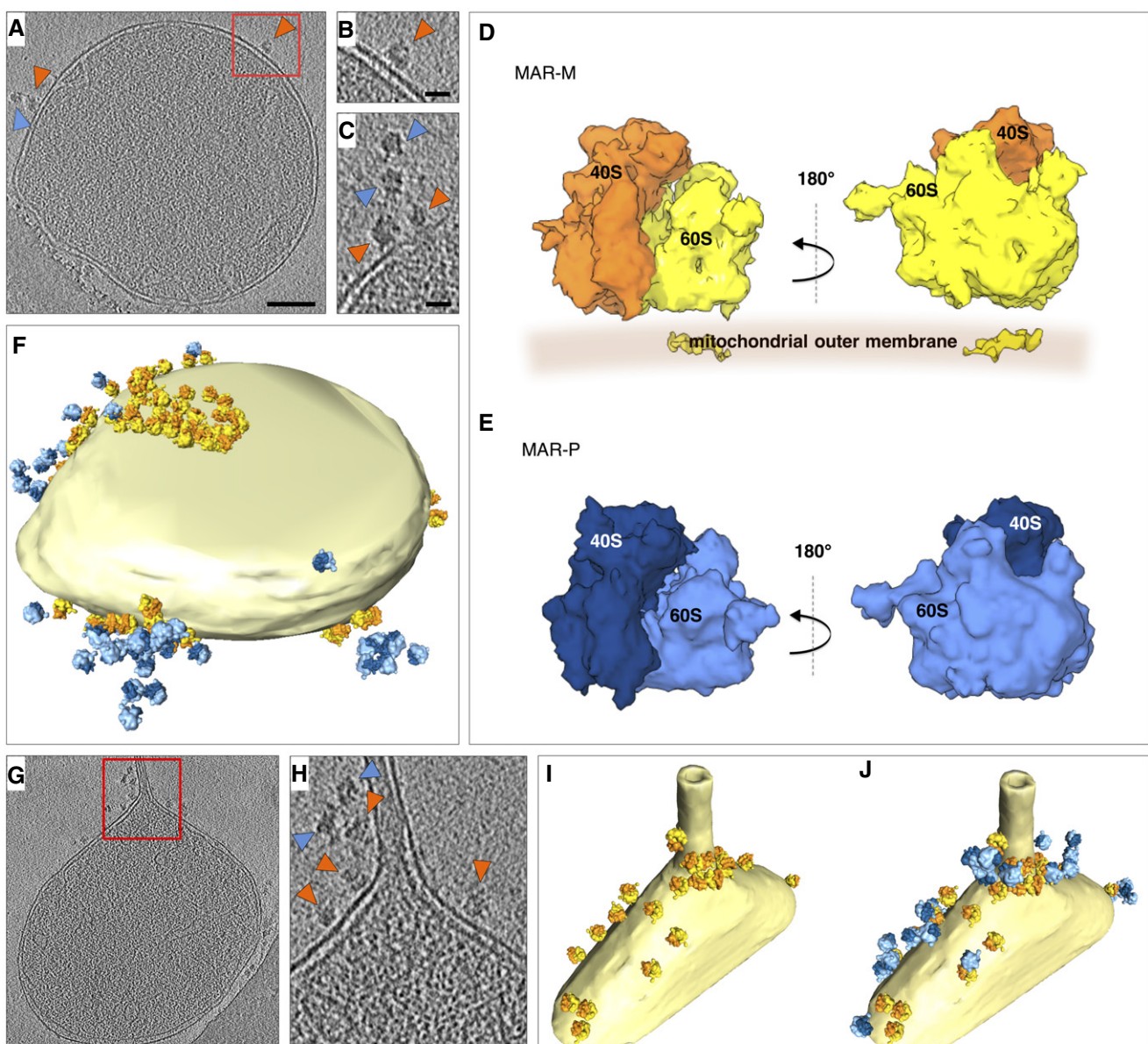

**Figure 3. Ribosomes form discrete clusters on mitochondria.**

A    Tomographic slice showing the location of ribosomes (MAR-M, orange arrowheads; MAR-P, blue arrowheads), associated with a mitochondrion. Scale bar, 0.1 μm.

B    Enlargement of the box shown in (A). Scale bar, 20 nm.

C    Tomographic slice showing the arrangement of MAR-M (orange arrowheads) and MAR-P (blue arrowheads). Scale bar, 20 nm.

D    StA of the MAR-M population (n = 1,215 subvolumes). The 60S subunit (yellow) and 40S subunit (orange) are shown with respect to the position of the mitochondrial membrane. The density shown within the membrane is attributable to the bilayer, not the TOM complex.

E    StA of the MAR-P population (n = 419 subvolumes). The 60S subunit (light blue) and 40S subunit (dark blue) are shown.

F    Surface-rendered mitochondrion as shown in (A), showing the distribution of MAR-M and MAR-P groups.

G    Tomographic slice showing the location of ribosomes bound to a mitochondrial outer membrane that has a partially tubular morphology. Scale bar, 0.1 μm.

H    Enlargement of the box shown in (G), showing the arrangement of MAR-M (orange arrowheads) and MAR-P (blue arrowheads). Scale bar, 20 nm.

I, J    Surface-rendered mitochondrion as shown in (H), showing the MAR-M distribution (I) and with the MAR-P group included (J).

(Figs 6G and EV5). The more notable difference was the variation in tethering distances, with variance calculated at 8.6 nm for MAR-M and 3.2 nm for ER-R populations, respectively (Figs 6G and H, and EV5). With respect to tethering distances, the ER-R group displayed a clear narrow distribution, with ~70% of ribosomes within the range 10–14 nm from the membrane. The MAR-M group, however,

displayed a much wider distribution, with only ~50% within the same range (Fig 6H). A StA calculated for the MAR-M population that included only ribosomes located within the 10–14 nm range (240 particles, a similar number to that used in the ER-R average) did not result in additional information (data not shown). Due to the low number of ribosomes bound in conditions without CHX

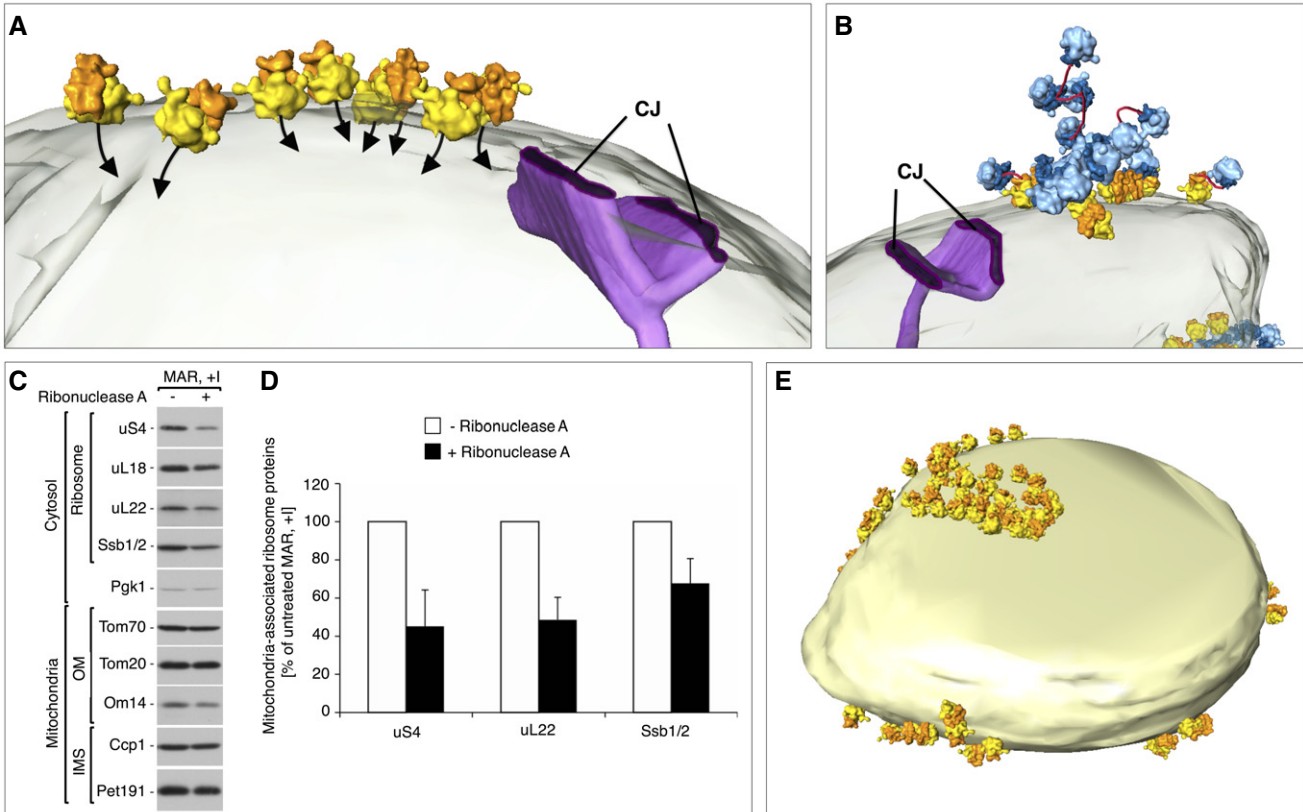

**Figure 4. Ribosomes are orientated for import on the mitochondrial surface.**

A  Enlargement of a MAR-M cluster from the mitochondrion shown in Fig 3F, depicting the position of the polypeptide exit tunnel (black arrows) with respect to the mitochondrial outer membrane (transparent) and a crista junction (CJ, purple).

B  A MAR-M cluster and associated MAR-P group are shown with respect to the mitochondrial outer membrane (transparent) and a crista junction (CJ, purple). The potential path of polysomal mRNA is shown (red).

C  The steady-state protein levels of gradient-purified MAR (MAR, +I) upon ribonuclease A treatment shows that the amounts of ribosomal proteins and Ssb1/2 are reduced compared to the non-digested sample. Samples were analyzed by SDS–PAGE followed by immunodecoration with specific antisera. OM, outer membrane; IMS, intermembrane space.

D  Quantification of the ribosomal protein levels from untreated MAR and after treatment with ribonuclease A, as shown in (C). The protein levels of uS4, uL22, and Ssb1/2 in untreated MAR were set to 100%. Data are presented as the mean ± SEM. *n* = 3 biological replicates.

E  Surface-rendered mitochondrion as shown in Fig 3F, showing the distribution of MAR-M only. The MAR-P population is not shown.

stabilization (Mg$^{2+}$ only), StA was not possible. Such a flexible mode of tethering agrees with the observation that the MAR-M population exhibits a significant degree of orientational flexibility with respect to the position of the polypeptide exit tunnel relative to the membrane (Fig 4A).

## Discussion

By cryoET of mitochondria isolated in the presence of Mg$^{2+}$, we were able to provide supportive evidence for the existence of co-translational import into isolated mitochondria. Using CHX-arrested RNCs, we performed StA and biochemical analyses to visualize ribosomes on the mitochondrial surface and to demonstrate that this is due to nascent chain import. This is based on several lines of evidence obtained in this study and is described as follows. Firstly, we were able to detect the ribosome–TOM complex interaction biochemically, which was reversible by induction of nascent chain

release. CryoET and StA revealed two groups of associated ribosomes, a distinct population located at the mitochondrial membrane (MAR-M), and a second group of soluble polysomes (MAR-P). The MAR-M group was directionally orientated with the polypeptide exit tunnel pointing toward the membrane for import and was tethered through the TOM complex by the polypeptide chain. The ribosomes in the MAR-P population displayed more undefined orientations. In human cells, polysomes were found to exist in various conformations, ranging from unordered to helical, planar, and spiral [59]. It is possible that organelle isolation and thus the absence of certain cytosolic factors could result in the predominantly undefined orientations described here.

The tethering distance between MAR-M and the mitochondrial membrane and ER-R and the ER membrane is calculated as 12–13 nm, but the variance is ~threefold more for MAR-M (8.6 nm compared to 3.2 nm). As both populations have been CHX-treated identically, we thus deduce that the larger variation in tethering distance in the MAR-M group is likely due to the flexibility and

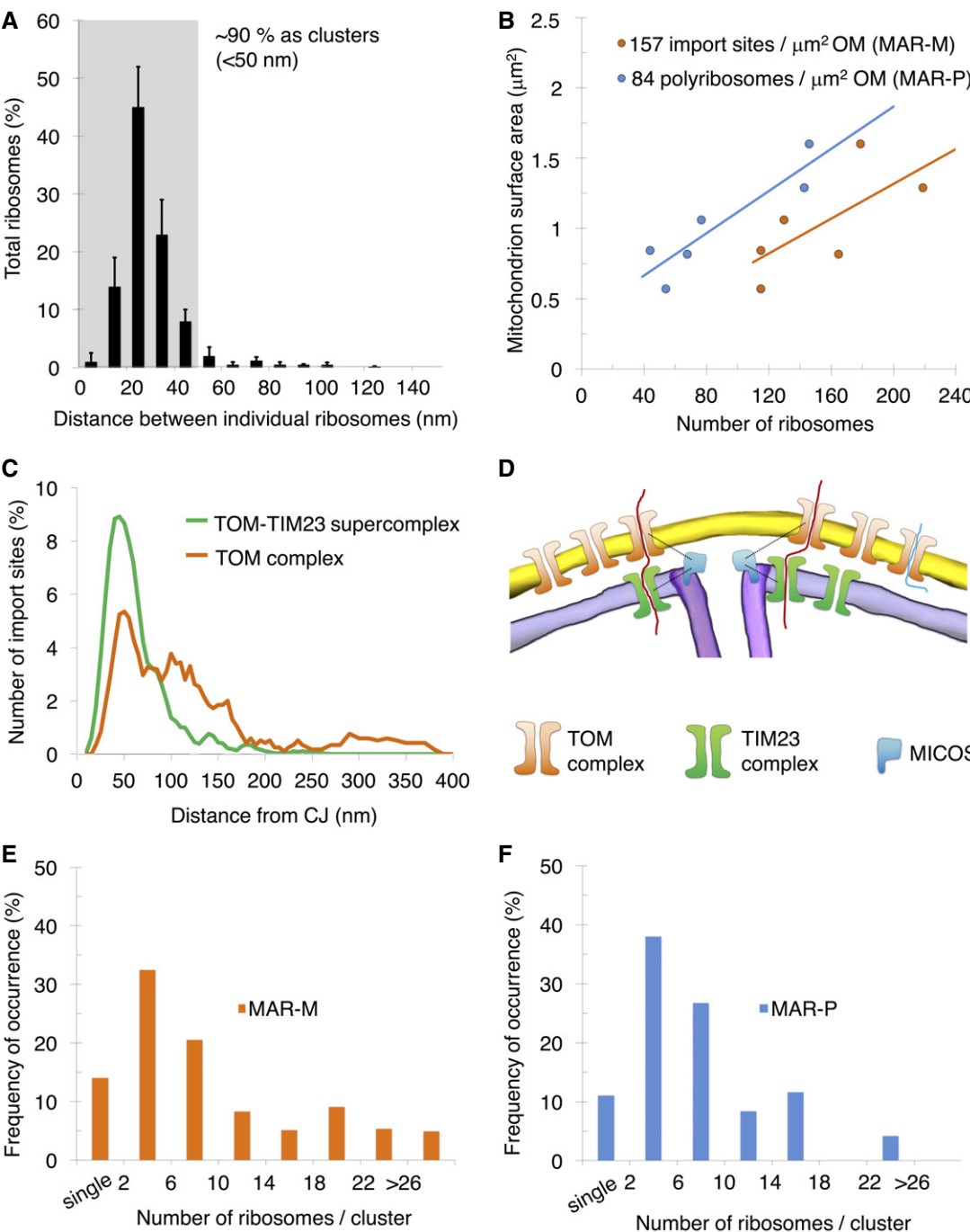

**Figure 5.  Ribosomes bind to mitochondria in discrete clusters near CJs.**

A   Histogram showing closest-neighbor distribution distances between individual ribosomes in the MAR-M group, expressed in percent. Error bars indicate the standard deviation of the frequency distribution for each minimal distance. *n* = 6 mitochondria (910 ribosomes), combined from two independent sample preparations.

B   Scatter plot showing the number of ribosomes (MAR-M, orange; MAR-P, blue) correlated with the surface area of individual mitochondria.

C   Distribution plot showing the number of import sites (expressed in percent) measured for TOM-TIM23 supercomplexes (green, *n* = 9 mitochondria (836 import sites)) and ribosome-labeled TOM complexes (orange, *n* = 6 mitochondria (397 ribosomes)), correlated with their distance from the nearest CJ. Data are plotted as a moving average in order to reduce the appearance of short-term fluctuations.

D   Schematic showing the distribution of TOM and TIM23 complexes in the mitochondrial membranes based on data shown in (C). The mitochondrial contact site and cristae organizing system (MICOS), responsible for formation and maintenance of the crista junction, is shown with respect to the TOM and TIM23 complexes.

E   Histogram showing the number of MAR-M per cluster expressed in percent. *n* = 6 mitochondria (923 ribosomes), combined from two independent sample preparations.

F   Histogram showing the number of MAR-P per cluster expressed in percent. *n* = 6 mitochondria (532 ribosomes), combined from two independent sample preparations.

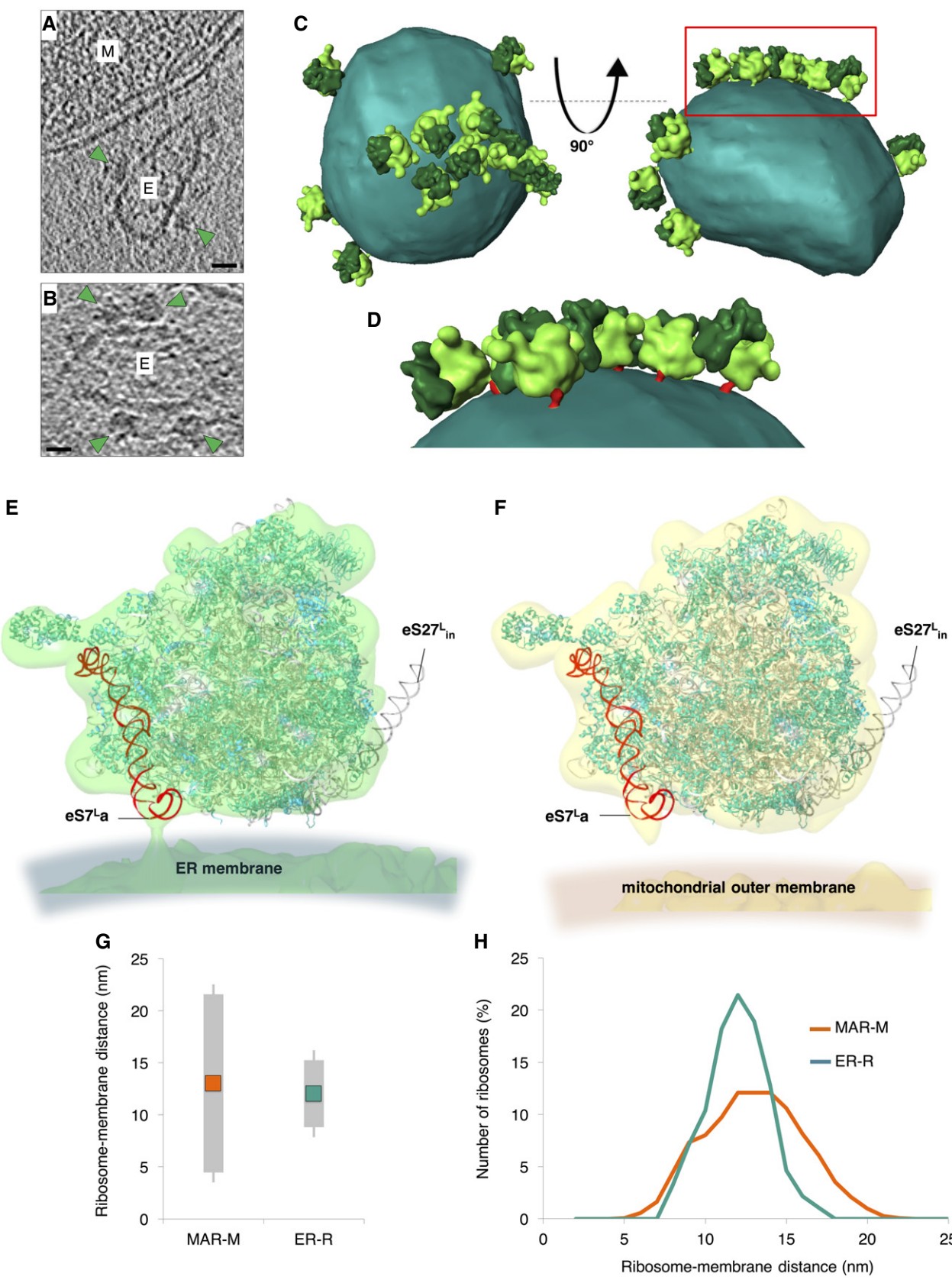

**Figure 6.**

**Figure 6.   Ribosomes are tethered to mitochondria through the strength of the nascent chain interaction.**

A, B   Reconstructed tomographic slices showing the location of ribosomes (green arrowheads) bound to rough ER membrane vesicles (marked E) that co-purify with mitochondria (marked M). Scale bars, 20 nm.

C   Surface-rendered rough ER membrane (sea green) showing the position of associated ER-R (60S bright green/40S dark green). $n$ = 230 subvolumes.

D   Enlargement of the box shown in (C). ER-R attachment to the membrane via eS7$^L$a is shown (red).

E, F   Docked X-ray structures show the positions of ribosomal proteins (teal) and rRNA (gray) in comparably filtered StAs of ER-R (green) and MAR-M (yellow) structures. eS7$^L$a (red) and eS27$^L_{in}$ (black) are also shown.

G   Graph showing the average distance between MAR-M and the mitochondrial outer membrane (orange) and ER-R and the ER membrane (teal), and the corresponding variance of tethering distances (gray bars). Calculations are made from the base of the cleft between the 60S and 40S subunits. $n$ = 15 mitochondria/tomograms (MAR-M) and $n$ = 11 tomograms (ER-R), accruing 964 ribosomes in total from five independent sample preparations.

H   Distribution plot showing the number of ribosomes (expressed in percent) in MAR-M (orange) and ER-R (teal) data sets, correlated with their distance from the membrane. Data are plotted as a moving average in order to reduce the appearance of short-term fluctuations.

varying angle of attachment afforded by the connection made through a nascent polypeptide chain and not any additional stabilizing partners. This does not exclude the presence of a specific receptor for ribosomes on mitochondria that may be required for earlier steps of import, such as binding and initiation, nor under conditions without CHX treatment. Interestingly, however, mRNA digestion was sufficient to dissociate ~50% of ribosomes from the mitochondrial surface. In addition, dissipation of the membrane potential by the chemical uncoupler CCCP affected ribosome association with mitochondria only if CCCP treatment preceded the addition of CHX. This indicates that post-lysis RNC recruitment to mitochondria did not have a significant effect on our results. In our ER-R StA, a connection is observed between the ribosome and the membrane by eS7$^L$a, which is known to be flexible in yeast [66]. This could explain why eS7$^L$a appears to be partially twisted away in both structures determined here, similar to that observed previously [4].

By cryoET and StA in this study, we were able to locate 167 TOM complexes per μm$^2$ outer membrane surface, ~twofold more than the 69 TOM-TIM23 supercomplex import sites determined previously [40]. This is in agreement with the fact that TOM is more abundant in mitochondria than TIM23 [67]. By directly comparing the two data sets, we also demonstrate that import through the TOM complex occurs in the vicinity of CJs, but this distribution is significantly broader than for arrested TOM-TIM23 supercomplexes. Our data therefore highlight key roles that the TIM23 complex may play in the mitochondrial organizing network. Proteins imported by the TOM-TIM23 route were previously observed to form clusters, also around fusion sites [40]. This is in agreement with the distribution of ribosome–TOM clusters observed here and also with respect to a potential fission constriction. Yeast proteins that are reportedly involved in fusion and fission are imported to mitochondria from cytosolic ribosomes [57,68]; thus, our data are consistent with the idea that import sites can redistribute to specific regions of mitochondria [40].

In conclusion, our data provide structural evidence supporting the theory that nuclear-encoded mitochondrial proteins can be synthesized locally at the mitochondrial outer membrane. Most likely, localized translation is initiated by mRNA recruitment to the mitochondrial surface [23,28,69]. Then, during ongoing translation the distance between the nascent chain and protein translocase is short, increasing the import efficiency [70]. Assuming that protein translocation is much faster than protein translation [71], nascent chain length may determine the time span that ribosomes are in contact with the mitochondrial surface in the manner of our characterized MAR-M population. It is therefore no surprise that the most studied protein thought to be delivered to mitochondria

in a co-translational manner is Fum1, with a larger than average molecular weight [72]. By stalling translation with CHX, we could observe different ribosome populations, including strings of polysomes present on the mitochondrial surface. Thus, at any given time, only a fraction of ribosomes are seen to interact with the TOM complex, while translating a single mRNA molecule.

Correct mRNA and protein delivery are likely more challenging with increasing cell volume and a higher demand for timely organization of mitochondrial biogenesis [73]. An interesting case was recently reported for the MDI A-kinase anchor protein, present in the mitochondrial outer membrane. MDI recruits a translation stimulator, La-related protein, and promotes mRNA tethering and local protein translation during oogenesis and early embryonic development of *Drosophila melanogaster* [74]. MDI-La-related protein complex formation was crucial for successful hatching and mitochondrial DNA replication, pinpointing the requirement for mRNA localization in efficient mitochondrial biogenesis. Thus, the importance of recruiting RNA molecules coding for mitochondrial proteins to the outer membrane and their localized translation is likely enhanced in specific cell types and developmental stages of higher eukaryotes.

## Materials and Methods

### Yeast strains and growth conditions

The strains used in this study were derivatives of *Saccharomyces cerevisiae* YPH499 (MATa, ade2-101, his3-Δ200, leu2-Δ1, ura3-52, trp1-Δ63, lys2-801) or BY4741 (MATa, his3Δ1; leu2Δ0; met15Δ0; ura3Δ0). The YPH499 strains carrying centromeric plasmids that express Tom40, Tom40$_{HA}$, or Tom22$_{HIS}$ were described previously [75–77]. A strain that carried chromosomally integrated uL13$_{TAP}$ (YIL133C) was purchased from GE Dharmacon (Lafayette, CO, USA).

Yeast were grown at 19–24°C on YPGal medium (1% w/v yeast extract, 2% w/v bactopeptone, 2% w/v galactose) with the addition of 0.1% w/v glucose or YPG medium (1% w/v yeast extract, 2% w/v bactopeptone, 3% w/v glycerol) to mid-logarithmic phase. To stabilize ribosomes, media were supplemented with 50 μg/ml of CHX for the final 45 min of the culture as indicated.

### Purification of mitochondria and MAR samples

Crude mitochondria were isolated according to a standard procedure [78] and resuspended in sucrose/MOPS (SM) buffer composed of

250 mM sucrose, 10 mM MOPS-KOH, pH 7.2. For crude MAR isolation, solutions were supplemented with 50 μg/ml CHX and 2 mM Mg(OAc)$_2$. For protein steady-state level analysis, mitochondria were solubilized in Laemmli buffer with 50 mM DTT, denatured at 65°C for 15 min, and analyzed by SDS–PAGE and Western blotting.

For further MAR purification, OptiPrep iodixanol density gradient medium (Sigma-Aldrich, St. Louis, MO, USA) was used. Crude MAR were separated on a step gradient with 10 layers (1 ml volume each) ranging from 0 to 27% v/v of iodixanol in gradient buffer (10 mM Tris–HCl, 8.75% w/v sorbitol, 2 mM Mg(OAc)$_2$, 50 μg/ml CHX, pH 7.4) by centrifugation at 80,000 × g for 40 min at 4°C using a SW41 Ti rotor (Beckman Coulter Inc., Miami, FL, USA). To analyze the organellar sedimentation profile, each gradient fraction was collected and precipitated with 10% (w/v) trichloroacetic acid (Carl Roth GmbH). The protein pellet was washed with ice-cold acetone, solubilized in urea sample buffer (6 M urea, 125 mM Tris–HCl, 6% SDS, 50 mM DTT and 0.01% (w/v) bromophenol blue, pH 6.8), denatured at 37°C for 15 min and analyzed by SDS–PAGE followed by Western blotting. For cryoET analysis, fractions with the highest mitochondrial content (corresponding to 15–21% iodixanol concentrations) were pooled, diluted 10-fold with SM buffer supplemented with 50 μg/ml CHX and 2 mM Mg(OAc)$_2$ and centrifuged at 22,000 × g for 20 min at 4°C to re-isolate MAR. Pelleted MAR were resuspended in SM buffer as before and used for further analysis.

## Isolation of high molecular weight membranes

To isolate HMW membranes, yeast cells were harvested, washed with ice-cold water, and disrupted in Lysis buffer (20 mM Tris–HCl, 10% w/v glycerol, 100 mM NaCl, 2 mM PMSF, 50 mM iodoacetamide, pH 7.4) with glass beads (425–600 μm, Sigma-Aldrich) using a Cell Disruptor Genie (Scientific Industries, Bohemia, NY, USA) at 2,800 rpm for 7 min at 4°C. To isolate HMW membranes under ribosome stabilizing conditions, lysis buffer was supplemented with 2 mM Mg(OAc)$_2$ and 50 μg/ml CHX. Cell debris was removed by centrifugation at 4,000 × g for 5 min at 4°C. HMW membranes were pelleted by centrifugation at 20,000 × g for 15 min at 4°C, washed, and resuspended in lysis buffer. The protein concentration was determined by the Bradford method. To confirm mitochondrial enrichment, equal amounts of control mitochondria and HMW membranes, based on protein concentration, were solubilized in Laemmli buffer containing 50 mM DTT, denatured at 65°C for 15 min and protein levels were analyzed by SDS–PAGE and Western blotting.

## Dissipation of the mitochondrial inner membrane electrochemical potential

Cells were treated with 10 μM CCCP (Sigma-Aldrich) for 0.5–3 h before cell harvesting. Translation was inhibited by addition of 50 μg/ml CHX prior to cell harvesting and followed by MAR or HMW membranes isolation.

## Nascent chain release assay

In order to analyze ribosome dissociation from mitochondria upon nascent chain release, 55 μg of crude mitochondria or MAR were suspended in 550 μl of release buffer (10 mM HEPES, 250 mM sucrose, 80 mM KCl, 5 mM MgCl$_2$, 5 mM methionine, 10 mM KH$_2$PO$_4$) or SM buffer, both supplemented with 0–1.5 M hydroxylamine (Sigma-Aldrich), 3 mM puromycin dihydrochloride (Sigma-Aldrich), or 25 mM EDTA, adjusted to pH 7.4 with HCl and incubated for 15 min at 30°C with gentle shaking. Mitochondria were re-isolated by centrifugation at 20,000 × g, washed with SM buffer and analyzed by SDS–PAGE followed by Western blotting. To purify MAR after nascent chain release, 2 mg of isolated crude MAR were incubated for 15 min at 30°C in 2 ml of release buffer with 1.5 M hydroxylamine and separated on 0–27% iodixanol gradient.

## Fragmentation of mitochondria-bound polysomes

1 mg of MAR was incubated with 50 μg/ml ribonuclease A from bovine pancreas (Sigma-Aldrich) for 15 min at 4°C with gentle shaking to digest mRNA. Next, ribonuclease-treated MAR were purified on a 0–27% iodixanol gradient to remove dissociated ribosomes and were analyzed by SDS–PAGE followed by Western blotting.

## Immuno-affinity purification of Tom40$_{HA}$

MAR (600 μg) or HMW membranes (3 mg) isolated from cells expressing either a wild-type or HA-tagged version of Tom40 were solubilized in digitonin buffer A (1% w/v digitonin, 20 mM Tris–HCl, 150 mM NaCl, 10% w/v glycerol, 50 mM iodoacetamide, 1 mM PMSF, 2 mM Mg(OAc)$_2$, 50 μg/ml CHX, pH 7.4) for 20 min at 4°C. After a clarifying centrifugation at 20,000 × g for 15 min at 4°C, supernatants were incubated with anti-HA agarose (Sigma-Aldrich) for 1.5 h at 4°C. Protein complexes were eluted by incubation with Laemmli buffer with 50 mM DTT. Samples were analyzed by SDS–PAGE and Western blotting.

## Immuno-affinity purification of Tom22$_{His}$

1 mg of isolated MAR containing HIS-tagged Tom22 (Tom22$_{HIS}$) was suspended in Buffer B (10 mM MOPS-KOH, 250 mM sucrose, 80 mM KCl, 5 mM MgCl$_2$, 5 mM methionine, 10 mM KH$_2$PO$_4$/K$_2$HPO$_4$, pH 7.2) supplemented with 25 mM EDTA in order to disrupt ribosomes. Control samples were mixed with Buffer B without EDTA. After incubation for 20 min at 4°C, all samples were centrifuged at 20,000 × g for 10 min at 4°C, washed with Buffer B, and the pellet solubilized in Digitonin buffer C (1% w/v digitonin, 20 mM Tris–HCl, 100 mM NaCl, 10% w/v glycerol, 50 mM iodoacetamide, 20 mM imidazole, 1 mM PMSF, 2 mM Mg(OAc)$_2$, 50 μg/ml CHX, pH 7.4) for 20 min at 4°C. After a clarifying centrifugation at 20,000 × g for 15 min at 4°C, the supernatant was incubated with Ni-NTA agarose (Qiagen, Hilden, Germany) for 1 h at 4°C. Protein complexes were eluted by incubation with elution buffer (20 mM Tris–HCl, 100 mM NaCl, 400 mM imidazole, pH 7.4). Eluted proteins were precipitated with StrataClean resin (Agilent Technologies, Santa Clara, CA, USA). The samples were incubated with Laemmli buffer supplemented with 50 mM DTT at 65°C for 15 min and analyzed by SDS–PAGE followed by Western blotting.

## Immuno-affinity purification of uL13$_{TAP}$

uL13$_{TAP}$ cells were treated with CHX, pelleted, and washed with ice-cold water. Yeast cells were resuspended in lysis buffer

supplemented with 2 mM Mg(OAc)$_2$ and 50 μg/ml CHX, followed by disruption with glass beads using a Cell Disruptor Genie at 2,800 rpm for 7 min at 4°C. Cell debris were removed by centrifugation at 20,000 × $g$ for 15 min at 4°C. The protein concentration of the supernatant (cytoplasmic fraction) was determined by the Bradford method. 3 mg of protein was incubated with 1.5 M hydroxylamine for 30 min at 30°C with gentle shaking. Samples were cooled on ice and subjected to IgG–Sepharose (GE Healthcare) affinity chromatography for 1 h at 4°C. The column was washed three times with washing buffer (20 mM Tris–HCl, 150 mM NaCl, 2 mM Mg (OAc)$_2$, 50 μg/ml CHX, pH 7.4), followed by the elution of protein complexes with Laemmli buffer with 50 mM DTT. Samples were analyzed by SDS–PAGE and Western blotting.

### Generation of RNCs and release assay

[$^{35}$S] methionine labeled Tim9-RNCs were generated as described previously [54]. Radiolabeled RNCs were resuspended in release buffer supplemented with 1.5 M hydroxylamine and incubated for 30 min at 30°C with gentle shaking. Reaction mixtures were mixed with Laemmli buffer containing 50 mM iodoacetamide, denatured at 65°C for 15 min and analyzed by SDS–PAGE followed by autoradiography (Variable Mode Imager Typhoon Trio, GE Healthcare).

### Electron cryo-tomography

Mitochondrial samples at a protein concentration of ~5 mg/ml total mitochondrial protein were mixed 1:1 with 10 nm protein A-gold (Aurion, Wageningen, The Netherlands) as fiducial markers and applied to glow-discharged R2/2 Cu 300 mesh holey carbon-coated support grids (Quantifoil, Jena, Germany) by gentle pipetting. Grids were blotted for ~4 s in a humidified atmosphere and plunge-frozen in liquid ethane in a homemade device. Tomography was performed either using a Tecnai Polara, Titan Krios (FEI, Hillsboro, USA), or JEM-3200FSC (JEOL, Tokyo, Japan) microscope. All microscopes are equipped with field emission guns operating at 300 keV, K2 Summit direct electron detector cameras (Gatan, Pleasanton, USA), and either a post-column Quantum energy filter operated at a slit width of 20 eV (FEI microscopes) or an in-column energy filter operated with a slit width of 40 eV (JEOL microscope). Dose-fractionated tomograms (three to eight frames per projection image) were typically collected from +60° to −60° at tilt steps of 2° and 5–8 μm underfocus with a total dose per tomogram of < 140e$^-$/Å$^2$. Data were collected using Digital Micrograph (Gatan) with various pixel sizes (depending on the microscope) per image. Tomograms were aligned using the gold fiducial markers and volumes reconstructed by weighted back-projection using the IMOD software [79]. Contrast was enhanced by nonlinear anisotropic diffusion (NAD) filtering in IMOD [80]. Segmentation was performed using AMIRA (FEI).

### Subtomogram averaging

Data collected at a nominal magnification of 42,000× (corresponding to a pixel size of 3.3 Å) on the Titan Krios were used for all StA. For the MAR-M and ER-R populations, two-point co-ordinates corresponding to the center of the ribosome and the center of either the outer mitochondrial or ER membrane were marked manually in IMOD [79]. Subvolumes from twice-binned tomograms were

extracted from NAD filtered data and an initial alignment and averaging performed in SPIDER [81]. This average was used as a reference for alignment and refinement using PEET [82]. A full 360° search was performed in Phi (twist around the particle), whereas Theta and Psi (bending in the x–y plane and z angles, respectively) covered only +/−90°. 1,215 subvolumes were used for the MAR-M structure and 230 subvolumes for the ER-R calculation, using a mask to exclude the membrane from the alignment. In the final iteration step for the MAR-M average, NAD-filtered tomograms were replaced by unfiltered contrast transfer function (CTF)-corrected data. Due to the reduced particle number for the ER-R population, this final step was not performed. Resolution estimates were obtained using conventional "even/odd" Fourier shell correlation (FSC), applying the 0.5 FSC criterion and using a mask to exclude the membranes from this estimate. In order to visualize the distribution and orientation of the MAR-P population in 3D space, a StA was also calculated. One-point co-ordinates were selected in the center of each ribosome, and subvolumes extracted for a full angular search in all three directions. NAD-filtered tomograms were again replaced by unfiltered CTF-corrected data, in order to compare MAR-M and MAR-P structures. Large and small ribosomal subunits were segmented in Chimera (UCSF, San Francisco, USA) for display, which was also used to remove low contrast background noise using the "hide dust" tool. NAD-filtered ribosome populations were displayed on membranes using AMIRA. X-ray data of yeast ribosomes (PDB-4V6I with PDB-3IZD, including a model of the position of eS27$^L$) [66] were docked into comparably NAD-filtered 3D maps of MAR-M or ER-R structures using Chimera. StAs have been uploaded to the EMDB under the accession numbers EMD-3762 (MAR-M), EMD-3763 (MAR-P), and EMD-3764 (ER-R).

### Calculation of the number of ribosomes associated with each mitochondrion

In order to calculate the approximate number of ribosomes bound to mitochondria during optimization of sample preparation (Fig 1A), only side-view ribosomes were counted. This is due to the "missing wedge" of information in tomography and the difficulty in identifying ribosomes bound to the upper and lower surfaces of mitochondria, especially those that are large and dense (> 500 nm). These values should therefore not be taken as absolute, but rather as a relative comparison between all four sample preparation conditions. Sample size is 22 mitochondria in total, which accumulate as follows: 30 MAR in +Mg(OAc)$_2$, 206 MAR in +Mg(OAc)$_2$ +CHX, and 824 MAR in +Mg(OAc)$_2$ +CHX +I. After further data collection, an accurate absolute value was calculated for MAR-M and MAR-P populations under final stabilizing conditions (+Mg(OAc)$_2$ +CHX +I in Fig 1A), by selecting only mitochondria in thin ice (< 500 nm) for the analysis, whereby ribosomes could be clearly defined around the entire circumference (Fig 5B). This was performed for 923 MAR-M and 523 MAR-P data points, combined from six mitochondria. Calculation of mitochondrial surface area was performed as previously described [40].

### Calculation of ribosome distribution and clustering

The distance between ribosomes, and between ribosomes and CJs, was determined with a MATLAB (Mathworks, California, USA)

script as previously described [40]. In order to calculate an accurate value based on coverage of the entire mitochondrial surface, again only mitochondria that demonstrated both side-views and clear upper and lower surface views of ribosomes were included in the clustering analysis. This was performed for 923 MAR-M and 532 MAR-P, combined from six mitochondria. CJs cannot be resolved on the upper and lower mitochondrial surfaces by tomography; thus, data were collected for 397 MAR-M for the CJ analysis. Averaged histograms were calculated to depict the mean frequency of occurrence for each minimal distance. To account for the different numbers of ribosomes in each data set, the mean frequency was calculated as a percentage.

### Calculation of ribosome distances from membranes

To calculate the distance between MAR-M or ER-R and their respective membranes, the xyz co-ordinates corresponding to the position of the cleft between the 60S and 40S ribosomal subunits and the membrane were extracted and plotted. Again, only side-views of ribosomes were used due to the difficulty in accurately defining both the position of the cleft and the membrane in upper and lower surface views. The cleft was chosen as a reference point as it is a clearly definable feature in individual tomograms. This accrued 824 data points from 15 tomograms for MAR-P and 140 data points from 11 tomograms for ER-R.

### Miscellaneous

Protein concentration was measured by Bradford method using Roti-Quant (Carl Roth GmbH) with bovine serum albumin as a standard. SDS–PAGE was performed according to standard procedures. Protein extracts were examined on 12 and 15% acrylamide gels. Western blot was performed using PVDF membranes (Millipore, Billerica, MA, USA), and specific antisera were used for protein immunodetection. HA-tagged and TAP-tagged proteins were detected by the use of monoclonal anti-HA and PAP soluble complex antibodies (Sigma-Aldrich), respectively. Enhanced chemiluminescence signals were detected by X-ray films (Foma Bohemia, Hradec Kralove, Czech Republic), digitalized by Perfection V850 Pro scanner (EPSON, Long Beach, CA, USA) and quantified using ImageQuant TL (GE Healthcare) software. The images were processed using Photoshop CS4 (Adobe Systems, San Jose, CA, USA). The nomenclature of proteins is according to the Saccharomyces Genome Database (SGD). For ribosomal proteins, unified nomenclature was used according to [83].

**Expanded View** for this article is available online.

### Acknowledgements

We thank Werner Kühlbrandt for his support, Deryck Mills for excellent maintenance of the EM facility, Inmaculada Mora Espi and Magdalena Dlugolecka for experimental assistance, Sabine Rospert, Paulina Sakowska, and Sean Connell for materials and helpful advice, and Nikolaus Pfanner, Martin van der Laan, and Raffaele Ieva who participated in the published work on TOM-TIM23 supercomplexes shown in Fig 5C. This work was supported by the Max Planck Society, University of Exeter, Foundation for Polish Science—Welcome Programme co-financed by the EU within the European Regional Development Fund and National Science Centre, Poland (NCN), grant DEC-2013/11/B/NZ3/00974. P.B. was supported by NCN grant DEC-2013/11/D/NZ1/02294.

## Author contributions

VAMG and AC designed the study. VAMG, PC, and PB performed the experiments and evaluated the data together with AC. VAMG and PC prepared the figures. VAMG, PC, and AC wrote the manuscript. All authors commented on the manuscript.

## Conflict of interest

The authors declare that they have no conflict of interest.

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
