## [Review Process File · EMBO Reports]

Manuscript EMBO-2017-44261

Visualization of cytosolic ribosomes on the surface of mitochondria by electron cryo-tomography

Vicki A. M. Gold, Piotr Chroscicki, Piotr Bragoszewski, and Agnieszka Chacinska

Corresponding author: Vicki Gold, Living Systems Institute; Agnieszka Chacinska, The International Institute of Molecular and Cell Biology, University of Warsaw

Review timeline:

Submission date:	21 March 2017
Editorial Decision:	03 May 2017
Revision received:	20 June 2017
Editorial Decision:	07 July 2017
Revision received:	11 July 2017
Accepted:	14 July 2017

Editor: Esther Schnapp

Transaction Report:

1st Editorial Decision

03 May 2017

Thank you for your patience while your manuscript was peer-reviewed at EMBO reports. I apologize for the delay in getting back to you; we have only now received the full set of referee reports (pasted below) as well as referee cross-comments.

As you will see, the referees acknowledge that the findings are interesting. Referees 1 and 2 further agree that a very short CHX treatment and cryo-ET data to demonstrate a direct interaction between ribosomes and the TOM complex are beyond the scope of this study. However, they also agree that the long-term CHX treatment creates ambiguity regarding the biological relevance of the imaged intermediates, and that this caveat needs to be clearly articulated in the revised manuscript text. If it is feasible, major comment 2 of referee 1 could be addressed, and the text and data interpretation should be amended as suggested by the referees.

We would thus like to invite you to revise your manuscript with the understanding that the referee concerns must be fully addressed and their suggestions taken on board. Please address all referee concerns in a complete point-by-point response. Acceptance of the manuscript will depend on a positive outcome of a second round of review. It is EMBO reports policy to allow a single round of revision only and acceptance or rejection of the manuscript will therefore depend on the completeness of your responses included in the next, final version of the manuscript.

REFeree REPORTS

Referee #1:

1. Does this manuscript report a single key finding? YES/NO

YES. The manuscript describes structures of Ribosome-Nascent Chain Complexes associated with mitochondria. This is not a single key finding per se, but the manuscript reports a coherent and focused set of data.

2. Is the reported work of significance(YES), or does it describe a confirmatory finding or one that has already been documented using other methods or in other organisms, etc (NO)? YES/NO
NO. From the biological perspective, the reported results are largely confirmatory. The study in many ways parallels one of the author's previous work [doi:10.1038/ncomms5129], but essentially using stalled ribosomes instead of quantum dots to locate translocating peptides.

3. Is it of general interest to the molecular biology community? YES/NO
YES. Both the subject of mitochondrial protein import and the method of electron cryo-tomography should be of general interest to the molecular biology community.

4. Is the single major finding robustly documented using independent lines of experimental evidence (YES), or is it really just a preliminary report requiring significant further data to become convincing, and thus more suited to a longer format article (NO)? YES/NO
YES. The major finding is reasonably supported and in line with expectations from previous studies, however I have reservations about the physiological relevance of this finding.

Report:

In this manuscript, Gold et al provide a structural description of cytosolic ribosomes associated with mitochondria using electron cryo-tomography (cryoET). The major finding reported in this manuscript is that polysomes are found in discrete clusters, frequently near cristae junctions, sometimes with ribosomes that extend away from the surface of the outer membrane. The authors have recently used cryoET to characterize the sites of protein import using stalled translocation intermediates; this work is the logical extension, building upon previous observations that mRNAs of nuclear-encoded mitochondrial proteins are localized to and translated at the mitochondrial outer membrane.

The major weakness of this study is technical in origin and stems from the pervasive use of cycloheximide (CHX)-arrest to isolate mitochondria with associated ribosomes (MARs). This appears to have been an experimental necessity to stabilize the species of interest, but dramatically changing the kinetics of import may have important ramifications for the physiological relevance of the structures presented here, as is well documented in prior experiments using CHX to interrogate mRNA association with mitochondria by negative stain EM and by expression profiling [doi:10.1083/jcb.65.1.1; doi:10.1093/embo-reports/kvf025; doi:10.1126/science.1257522].

With those reservations in mind, the authors report the first three-dimensional view of RNCs at the mitochondrial outer membrane, making this the state of the art in our understanding of how cytosolic ribosomes can interact with mitochondria. The results largely confirm expectations; i.e. that the orientation of the ribosome supports nascent chain translocation, that the sites of translocation are clustered as the authors previously showed using a post-translational substrate trap [doi:10.1038/ncomms5129] and there is no evidence for a ribosome receptor (along with a broader distribution of distances to the membrane than observed for the rough ER). Discerning the physiological relevance of these observations awaits application of in situ cryoET [e.g. doi:10.1126/science.aad8857], beyond the scope of this work.

This manuscript in its current form provides valuable information for those interested in mitochondrial protein import, but for the broader audience there is probably little that will change the way they think about this important aspect of cell biology. The work necessary to address the physiological relevance of the intermediates studied here is beyond what is reasonable to request in a revision.

Major comments:

1. Given that translocation is faster than translation, and translation occurs on the order of a minute, the timescales of drug treatments are excessively long (e.g. 30-45 minutes with CHX, 1-3 hour for CCCP). If the intention was to preserve a snapshot of the in vivo state, CHX should have been

applied for approximately a minute. The use of long drug treatments throughout this study is a significant confounder to the physiological relevance of the results. Even the shortest duration of uncoupling, 30 minutes prior to CHX, under non-fermentable growth conditions is likely to impinge on translation in indirect ways.

2. The authors should consider characterizing MARs digested with ribonuclease to sever polysomes into single ribosomes. Those not directly in contact with the mitochondria should be removed during purification. This would be helpful in interpreting the clustering of MAR-Ms and the nature of MAR-Ps (if still present). Nuclease digests would also help with interpreting the presence of Ssb1 in TOM pulldowns and Egd1 in HMW membrane fractions.

Minor comments:

3. I was confused by the following statement "The purification step did not adversely affect the number of ribosomes bound stably to the outer membranes of mitochondria (Fig 1A & Fig EV1A)" (Pp 6, line 145-147) in light of the previous statement "These background ribosomes were often found in close proximity to mitochondria and made accurate statistical and structural analysis extremely challenging" (Pp 6, lines 137-139). If it was difficult to make an accurate measurement in the crude prep, comparing this value to the more accurate iodixanol purified fraction seems unnecessary.

4. The use of "steady state" in describing the levels of proteins is confusing since it implies the existence of kinetic data elsewhere, which I did not find.

Lastly, please note that I do not have the background necessary to assess the technical aspects of the cryoET.

Referee #2:

Gold et al. addressed a major question in the field of mitochondrial protein biogenesis, the association of cytosolic ribosomes with the surface of mitochondria. The manuscript presents impressive electron cryo-tomographic images combined with a statistical analysis and biochemical characterization. Mitochondria-associated ribosomes were typically observed in clusters. For the first time, ribosomes were found to be specifically oriented with the polypeptide exit tunnel pointing towards the mitochondrial outer membrane. In addition, ribosome clusters were often observed in the vicinity of crista junctions of the mitochondrial inner membrane. Gold et al. show that active TOM complexes display a broader distribution than active TOM-TIM23 presequence import sites, emphasizing the role of TOM as major mitochondrial protein import gate for various classes of precursor proteins. A further highlight of the manuscript is a comparison between mitochondria and endoplasmic reticulum (ER) associated ribosomes. The average distance between ribosomes and the membrane surface was comparable for ER and mitochondria, however, a much larger variation was observed for mitochondria associated ribosomes, pointing to a larger flexibility of the mitochondrial import system.

This is an important study that represents a major advance for our understanding of mitochondrial protein biogenesis. The authors present a careful biochemical analysis, followed by electron cryo-tomographic images of excellent quality.

Minor points:

1) The first sentence of the Abstract sounds a bit technical. I recommend to include an additional introductory sentence to explain the scope of the study.

2) The Results section describing Figures 1 and 2 is quite technical in some parts. I recommend some re-wording such that the main points will be more clearly visible.

3) The authors report the interesting observation that an increased accumulation of cytosolic ribosomes at the mitochondrial surface leads to the accumulation of a subunit of NAC at mitochondria. The authors should provide some background on NAC for the general readership and

spell out the name when first used.

Referee #3:

This paper addresses the question of whether there is co-translational transport of proteins into the mitochondria using biochemical and cryo-ET methods.

The authors isolated native mitochondria with bound translation arrested ribosomes. They show biochemically that these ribosomes translationally arrested using cycloheximide are associated to the TOM complex. Unfortunately, the authors do not show nor demonstrate their association or proximity by cryo-ET, only that the ribosomes cluster close to the mitochondrial crista junctions. The evidence for this clustering is sound but could be better represented in the figures of segmented mitochondria/ribosomes.

The authors provide good cryo-ET evidence that the interaction between ribosomes and mitochondria involves the nascent chain translocating through the mitochondrial membrane. Somewhat contradicting their biochemical data, cryo-ET does not show any specific interaction between ribosomes and the TOM complex, only proximity to its purported location near the crista junctions. Clustering of the TOM complexes is inferred from the clustering of membrane-bound ribosomes.

Not having studied this subject, I cannot judge the novelty of the finding and will leave this to the experts in this field. I can only say that the structural data provided supports the hypothesis that nuclear-encoded mitochondrial proteins are synthesized locally at the mitochondrial outer membrane. The mechanism of this recruitment remains to be elucidated.

The findings are well worth being published in EMBO reports. I would only suggest the authors to clarify a few paragraphs. In particular, in a number of places it is not clear from the wording if conclusions are drawn from the structural or biochemical data. Also, the conclusion is very much unfocused and the discrepancy between what was inferred from the new data and what comes from previous literature is not clear.

1st Revision - authors' response

20 June 2017

Regarding the general comment on the long-term treatment with CHX, we do appreciate the concerns of reviewers but would like to reiterate that this step was absolutely necessary and has provided the first view of ribosomes on the mitochondrial surface in 3D. To counter this concern, we have more clearly articulated the use of CHX with respect to kinetics and protein targeting in both the Results (pg.5 & 8) and Discussion sections (pg. 14).

Regarding data quantification, in all figures that originate from Western blot data (Fig 2D, 2F, 4D and EV2G) n = the number of biological replicates. In the remaining figures that contain numerical data obtained through cryoET, the wording has been altered to read n = number of mitochondria (and where appropriate corresponding number of ribosomes), which is equivalent to number of experiments. For subtomogram averaging, n = number of particles. Detailed explanations are given in the Materials and Methods section and in the legends. Error bars are described as SEM in all figures with two exceptions. In Figure 5A, a more appropriate measure of error was calculated, which is described in the legend. In order to depict differences between the two data sets shown in Figure 6G, the errors are described as variance in the legend. Scale bars are included in all microscopy images. Scale bars are not included in tomographic surface representations, where a 2D scale bar would not be appropriate to represent a 3D image.

POINT-BY-POINT RESPONSE: Included on the following pages.

Point by point response

We thank the Referees for their comments.

Referee 1, general comments:

The major weakness of this study is technical in origin and stems from the pervasive use of cycloheximide (CHX)-arrest to isolate mitochondria with associated ribosomes (MARs). This appears to have been an experimental necessity to stabilize the species of interest, but dramatically changing the kinetics of import may have important ramifications for the physiological relevance of the structures presented here, as is well documented in prior experiments using CHX to interrogate mRNA association with mitochondria by negative stain EM and by expression profiling [doi: 10.1083/jcb.65.1.1; doi:10.1093/embo-reports/kvf025; doi:10.1126/science.1257522].

We do appreciate that CHX will alter the kinetics of protein translation and import, but as described, it was a necessary step in order to be able to visualise sufficient numbers of ribosomes on the mitochondrial surface. We do not suggest in the manuscript that these CHX-stabilized ribosomes are importing proteins co-translationally, rather only that the Mg²⁺ stabilised ribosomes (without CHX) provide supportive evidence for its existence (lines 115-122). We also do not describe the kinetics of protein import, as this would indeed be difficult to prove from these data. Rather, we use the CHX-arrested ribosomes to locate the position of the TOM complex, which is not affected by the use of CHX. We have now included additional text in the Results (pg.5 & 8) and Discussion sections (pg. 14) to articulate more clearly why use of CHX was necessary, and the implications that this may have for the biological significance.

Major comments:

1. Given that translocation is faster than translation, and translation occurs on the order of a minute, the timescales of drug treatments are excessively long (e.g. 30-45 minutes with CHX, 1-3 hour for CCCP). If the intention was to preserve a snapshot of the in vivo state, CHX should have been applied for approximately a minute. The use of long drug treatments throughout this study is a significant confounder to the physiological relevance of the results. Even the shortest duration of uncoupling, 30 minutes prior to CHX, under non-fermentable growth conditions is likely to impinge on translation in indirect ways.

The time and dose of CHX treatment was optimized in order to achieve the highest enrichment of mitochondria bound-ribosomes, in order to determine the spatial distribution of the TOM complex in the outer membrane. Regarding CCCP, we show a time-dependent treatment in Fig 2C and D. We admit that 3h treatment with CCCP is long, but as shown in Fig 2C and 2D, we observe a dependence of ribosome association on the time of CCCP treatment. We thus avoided shorter treatments with CCCP followed by a long mitochondrial isolation procedure to prevent variation and thus ambiguity in the results.

2. The authors should consider characterizing MARs digested with ribonuclease to sever polysomes into single ribosomes. Those not directly in contact with the mitochondria should be removed during purification. This would be helpful in interpreting the clustering of MAR-Ms and the nature of MAR-Ps (if still present). Nuclease digests would also help with interpreting the presence of Ssb1 in TOM pulldowns and Egd1 in HMW membrane fractions.

This is an excellent suggestion and we have performed additional experiments and included new data as Figure 4C & D. Treatment of MAR with ribonuclease A shows a ~50% reduction in the amount of ribosomes present in the sample, in agreement with our cryoET data reporting on the absolute numbers of MAR-M and MAR-P populations (now shown in Figure 5B). In the same assay, Ssb1 was reduced up to ~70%, suggesting that it binds with higher affinity to MAR-M than to MAR-P. We have now added new discussion in the revised version to say that MAR-P ribosomes that have just initiated translation do not stably interact

with Ssb1, likely because the nascent chain is too short to emerge from the ribosomal exit tunnel. This is in agreement with literature and our data shown in Fig 2G.

Minor comments:

3. *I was confused by the following statement "The purification step did not adversely affect the number of ribosomes bound stably to the outer membranes of mitochondria (Fig 1A & Fig EV1A)" (Pp 6, line 145-147) in light of the previous statement "These background ribosomes were often found in close proximity to mitochondria and made accurate statistical and structural analysis extremely challenging" (Pp 6, lines 137-139). If it was difficult to make an accurate measurement in the crude prep, comparing this value to the more accurate iodixanol purified fraction seems unnecessary.*

We have re-worded the paragraph at lines 138-152 to clarify this section.

4. *The use of "steady state" in describing the levels of proteins is confusing since it implies the existence of kinetic data elsewhere, which I did not find.*

The expression "steady state" is commonly used to describe protein abundance, therefore we prefer to use this term.

Referee #2, minor points:

1) *The first sentence of the Abstract sounds a bit technical. I recommend to include an additional introductory sentence to explain the scope of the study.*

We have modified the first two sentences of the abstract accordingly.

2) *The Results section describing Figures 1 and 2 is quite technical in some parts. I recommend some re-wording such that the main points will be more clearly visible.*

The Results section has been re-worded to improve clarity and highlight the main points.

3) *The authors report the interesting observation that an increased accumulation of cytosolic ribosomes at the mitochondrial surface leads to the accumulation of a subunit of NAC at mitochondria. The authors should provide some background on NAC for the general readership and spell out the name when first used.*

The acronym has been defined and a background sentence on NAC had been included with an appropriate reference.

Referee #3, general comments:

The authors isolated native mitochondria with bound translation arrested ribosomes. They show biochemically that these ribosomes translationally arrested using cycloheximide are associated to the TOM complex. Unfortunately, the authors do not show nor demonstrate their association or proximity by cryo-ET, only that the ribosomes cluster close to the mitochondrial crista junctions. The evidence for this clustering is sound but could be better represented in the figures of segmented mitochondria/ribosomes.

We very much appreciate the recognition that our evidence for clustering is sound, and that our structural data supports our conclusions. Unfortunately, it was not possible to demonstrate ribosomal association with the TOM complex by cryoET and StA directly. This is because it is not yet possible to identify the TOM complex in the membrane, being as membranes scatter electrons more strongly than protein (meaning that identifying the proteins within membranes is very challenging). In some cases it has been possible to identify protein complexes in native membranes by cryoET and StA e.g. Sec61 in the ER membrane (Pfeffer *et al*,

2015, Nat Commun, doi:10.1038/ncomms9403). However, as mitochondria are significantly larger and more dense than ER membranes (limiting the resolution and ability to locate areas of interest), combined with our much lower number of particles available for StA per tomogram here, means that structural determination of the TOM complex is not yet possible by this technique. To improve the representation of clustering in figures, we have now included labels for the CJs in Figures 4A and B, and include additional data in the form of an Expanded View movie whereby an entire mitochondrion with ribosome clusters can be visualised in 3D.

The authors provide good cryo-ET evidence that the interaction between ribosomes and mitochondria involves the nascent chain translocating through the mitochondrial membrane. Somewhat contradicting their biochemical data, cryo-ET does not show any specific interaction between ribosomes and the TOM complex, only proximity to its purported location near the crista junctions. Clustering of the TOM complexes is inferred from the clustering of membrane-bound ribosomes.

In our opinion, the cryoET data does not contradict the biochemical data, but rather highlights a limitation of the methodology and technique at this time as described in the previous point. In fact, due to the fact that strikingly similar clusters of importing proteins are observed in this work (similar to that reported previously in Gold *et al*, 2014, Nat Commun; doi:10.1038/ncomms5129), our new cryoET data is in support of an interaction with the TOM complex, and is thus in complete agreement with our new biochemical data.

I would only suggest the authors to clarify a few paragraphs. In particular, in a number of places it is not clear from the wording if conclusions are drawn from the structural or biochemical data. Also, the conclusion is very much unfocused and the discrepancy between what with inferred from the new data and what comes from previous literature is not clear.

We have improved the general wording of the manuscript so that it is clearer where conclusions are drawn from current rather than previous data, and structural rather than biochemical data.

Thank you for the submission of your revised manuscript to our journal. We have now received the enclosed reports from the referee that was asked to assess it. The referee still has a few more minor suggestions that I would like you to incorporate before we can proceed with the official acceptance of your manuscript. Please also address all referee comments in a point-by-point response, which can be part of the cover letter.

- Figure 1A, 5A, EV1A, EV2I, and EV4 do not mention the number of independently performed experiments the data is based on. Are these single experiments each? This needs to be mentioned please.
- The bottom row in figure 2A seems to have a splice in the middle of the gel. Can you please explain what this is?
- Please provide an ORCID ID for Agnieszka Chacinska. She needs to enter the ID in her profile page in our online manuscript submission system. The system does unfortunately not allow us to do this for you.
- Please change the movie callout in the manuscript text to "Movie EV1". Please cut the movie legend from the manuscript and zip it with the movie file and upload the zipped file.
- Please provide a short running title for the manuscript.

I would like to suggest a few changes to the title and abstract, which need to be written in the present tense: "High resolution cryoET data show clustered cytosolic ribosomes on the surface of mitochondria" (or similar, please modify as you think fits).

"We employed electron cryo-tomography to visualize cytosolic ribosomes on the surface of mitochondria. Translation-arrested ribosomes reveal the clustered organisation of the TOM complex, corroborating earlier reports of localized translation. Ribosomes are shown to interact specifically with the TOM complex and nascent chain binding is crucial for ribosome recruitment and stabilization. Ribosomes are bound to the membrane in discrete clusters, often in the vicinity of the crista junctions. This interaction highlights how protein synthesis may be coupled with transport, and provides insight into the spatial organization of cytosolic ribosomes on mitochondria at unprecedented resolution."

REFeree REPORT

Referee #1:

In their revision, the authors present one additional experiment suggested in the previous review and make minor changes and clarifications throughout as requested. These revisions improve the clarity of the manuscript and address most of the concerns raised during the initial review. Below are some comments for the authors' and editor's consideration.

The abstract states that their results highlight "the importance of spatial organization for efficient mitochondrial import" (line 34-36). I don't think the results presented here speak at all to importance or function. The data presented describe this spatial organization with unprecedented resolution, but no experiments were performed to probe the importance of this organization.

The "biogenesis of mitochondrial proteins" is more of a "process" than a "factor" (line 64).

The number of mitochondrial proteins in the cited paper is 750, not 1000 (line 66, reference 9). Please use the correct value.

"Their association" (line 84) implies that all mRNAs localized to mitochondria depend on Cop1, when the cited paper (reference 30) only demonstrates this for one mRNA (Oxa1). Please adjust to more accurately portray the available evidence.

The use of "likely" (line 125) implies uncertainty, even though the effects of CHX on translation and targeting to mitochondria are documented (e.g. reference 27). Please delete.

Are the Ssb1 antibodies used specific to Ssb1, or do they show cross reactivity to other Hsp70s (e.g. Ssa1)? This is important in the claim made beginning on line 257, that Ssb1 has higher affinity for

MAR-M ribosomes. If the antibody also detects Ssa family members, the modestly higher signal may be ribosome independent, since Ssa is known to interact with Tom70. Please clarify.

To address referee 3's point, I think it should be more clearly spelled out that MAR-M ribosomes are inferred to be present at TOM complexes based on the combination of biochemical data presented here and existing data, both from the authors' previous study (reference 39) and many others. It should be made very explicit that TOM is not directly observed in the tomograms based on the author's response to referee 3. However, I found this confusing since figure 3D depicts electron density in the membrane. Is this attributable to TOM?

Lastly, I understand that the CCCP data demonstrate a time-dependent decrease in MAR, but maintain that at the times assayed, this could be due to any number of secondary changes that have nothing to do with the electrochemical potential at the inner membrane. A sentence acknowledging that the changes could be secondary to dissipation of the electrochemical potential would be appropriate given that other perturbations to ETC were not tested.

2nd Revision - authors' response

11 July 2017

Editorial comments:

1) Figure 1A, 5A, EV1A, EV2I, and EV4 do not mention the number of independently performed experiments the data is based on. Are these single experiments each? This needs to be mentioned please.

These data all stem from cryoET analysis and we appreciate the difficulty in clarifying experimental repeats. In the previous version, the wording in the legend was altered to read n = number of mitochondria analysed (and where appropriate corresponding number of ribosomes), which is often more meaningful than the number of independent experiments, which cannot always be defined in the same way as biochemical experimental repeats. To try and provide additional detail in Figure 1A and EV1A, we have now included an approximate value for the combined number of independent samples that were prepared for cryoET. Unfortunately, this number cannot be accurately defined as we have analysed tens to hundreds of control mitochondria preparations in parallel over the years (where the number of ribosomes is always 0), compared to 5 independent mitochondrial preparations (for MAR, +I), where the number of ribosomes has a specific value. In the more specific analyses shown in Figure 5A, EV2I and EV4, a value has now been given. For consistency, the same information has also been included in Figures 5E, F & 6G. In Figure EV2I, we would like to point out that 5 independent replicates were performed for the MAR, +I samples as we needed to maximise the number of available ribosomes for sub-tomogram averaging. In the comparison (MAR, +NH₂OH, +I), only one sample was prepared as this experiment was performed later for the first revision of our manuscript. To obtain a comparable number of replicates would have taken several months and would have been unnecessary for the conclusions drawn.

2) The bottom row in figure 2A seems to have a splice in the middle of the gel. Can you please explain what this is?

The line visible in the bottom row of Figure 2A is an artefact present also on the original X-ray film, likely a scratch made by the developing machine. Please also find included with our submission the full scan of the original film - TIF file (a) digitalized by the V850 PRO scanner (Epson). In the preparation of Figure 2A the TIF file was opened using Photoshop (Adobe) and the relevant area was cropped out as indicated in the accompanying file (b) by the dashed line, and was saved as a new TIF file. Next, the cropped image was placed and embedded in the Adobe Illustrator file that contained assembled Figure 2A. The part of Figure 2A that corresponds to Pdi1 was processed in exactly the same way as for all other presented Western blot results. None of our figures have been spliced and the observed anomaly does not influence our findings, thus we hope that this figure panel will be acceptable.

3) Please provide an ORCID ID for Agnieszka Chacinska. She needs to enter the ID in her profile page in our online manuscript submission system. The system does unfortunately not allow us to do this for you.

Agnieszka Chacinska has provided her ORCID ID in the online submission system.

4) Please change the movie callout in the manuscript text to "Movie EV1". Please cut the movie legend from the manuscript and zip it with the movie file and upload the zipped file.

This has been done.

5) Please provide a short running title for the manuscript.

We propose the following running title: Mitochondria-associated ribosomes

6) I would like to suggest a few changes to the title and abstract that needs to be written in present tense: High resolution cryoET data show clustered cytosolic ribosomes on the surface of mitochondria (or similar, please modify as you think fits)

We have changed the title to "Visualization of cytosolic ribosomes on the surface of mitochondria by electron cryo-tomography." We prefer not to use the term "high-resolution" as this would usually mean visualisation of secondary structure such as protein alpha helicies (<10 Å).

We employed electron cryo-tomography to visualize cytosolic ribosomes on the surface of mitochondria. Translation-arrested ribosomes reveal the clustered organisation of the TOM complex, corroborating earlier reports of localized translation. Ribosomes are shown to interact specifically with the TOM complex and nascent chain binding is crucial for ribosome recruitment and stabilization. Ribosomes are bound to the membrane in discrete clusters, often in the vicinity of the crista junctions. This interaction highlights how protein synthesis may be coupled with transport, and provides insight into the spatial organization of cytosolic ribosomes on mitochondria at unprecedented resolution.

Thank you for the suggestions and edits to the abstract. We have accepted the rewording, with only an exception to the phrase "at unprecedented resolution," for the same reason as described for the suggested change to the title. Thus, the last sentence has been modified as follows: "This interaction highlights how protein synthesis may be coupled with transport, and provides insight into the spatial organization of cytosolic ribosomes on the mitochondrial outer membrane."

Referee #1:

8) The abstract states that their results highlight "the importance of spatial organization for efficient mitochondrial import" (line 34-36). I don't think the results presented here speak at all to importance or function. The data presented describe this spatial organization with unprecedented resolution, but no experiments were performed to probe the importance of this organization.

We have re-worded this sentence as requested (lines 46-49), without referring to "unprecedented resolution" for reasons described in point 6.

9) The "biogenesis of mitochondrial proteins" is more of a "process" than a "factor" (line 64).

This word has been changed as suggested.

10) The number of mitochondrial proteins in the cited paper is 750, not 1000 (line 66, reference 9). Please use the correct value.

The number of identified mitochondrial proteins in reference 9 is indeed 750. However, the authors underlined that this value was actually an underestimate \bar{n} covering ~90% of all mitochondrial proteins. Additionally, a recently published paper (Morgenstern, Stiller, Lubbert, Peikert et al., Cell Reports, 2017; DOI: 10.1016/j.celrep.2017.06.014) expanded the number of defined mitochondrial proteins up to 901. Therefore, we have changed the text accordingly and have added the new reference (line 66).

11) "Their association" (line 84) implies that all mRNAs localized to mitochondria depend on Cop1, when the cited paper (reference 30) only demonstrates this for one mRNA (Oxa1). Please adjust to

more accurately portray the available evidence.

The text has been re-worded to avoid generalization (lines 84-86).

12) The use of "likely" (line 125) implies uncertainty, even though the effects of CHX on translation and targeting to mitochondria are documented (e.g. reference 27). Please delete.

This word was deleted as suggested.

13) Are the Ssb1 antibodies used specific to Ssb1, or do they show cross reactivity to other Hsp70s (e.g. Ssa1)? This is important in the claim made beginning on line 257, that Ssb1 has higher affinity for MAR-M ribosomes. If the antibody also detects Ssa family members, the modestly higher signal may be ribosome independent, since Ssa is known to interact with Tom70. Please clarify.

*The polyclonal Ssb1 antibody was raised in rabbits by immunization with the peptide AEVGLKRVTKAMSSR. This peptide corresponds to the C-terminal part of Ssb1 and is required for its interaction with the ribosome (Hanebuth et al., Nat Commun, 2016; DOI: 10.1038/ncomms13695). The C-terminal region of Ssb1 is not conserved within the Hsp70 family, including with Ssa1 (Boorstein et al., J Mol Evol, 1994; PMID: 8151709). Therefore, the increased signal that we identify is ribosome dependent. However, Ssb1 does have a paralog, Ssb2, that arose from genome duplication in *S. cerevisiae*. Due to their extremely high amino acid sequence identity (99%), both forms of Ssb proteins are recognized by the same antibody. Thus, we have changed the protein name Ssb1 into Ssb1/2 in the text and figures (Fig. 2A, 2B, 2G, 4C, 4D, EV1B and EV2H). For consistency, we have also changed the protein name Ssa1 to Ssa1/2 in Figure 2G, as the proteins are also paralogs. This does not affect our findings.*

14) To address referee #3's point, I think it should be more clearly spelled out that MAR-M ribosomes are inferred to be present at TOM complexes based on the combination of biochemical data presented here and existing data, both from the authors' previous study (reference 39) and many others. It should be made very explicit that TOM is not directly observed in the tomograms based on the author's response to referee #3. However, I found this confusing since figure 3D depicts electron density in the membrane. Is this attributable to TOM?

We have included additional clarification in the results (lines 266-267), explaining that the ribosome could be used as a tag to mark the position of the TOM complex as a direct outcome of our biochemical data. We also refer to our previous study and the work of others throughout, and how this relates to the current study. The confusion likely predominates from the concern regarding direct visualization of TOM as the electron density seen in the membrane in our images. Therefore, additional information has been included in the legend of Figure 3D, stating that the density visible underneath the ribosome is attributable to the membrane, not the TOM complex.

15) Lastly, I understand that the CCCP data demonstrate a time-dependent decrease in MAR, but maintain that at the times assayed, this could be due to any number of secondary changes that have nothing to do with the electrochemical potential at the inner membrane. A sentence acknowledging that the changes could be secondary to dissipation of the electrochemical potential would be appropriate given that other perturbations to ETC were not tested.

We agree with the comment and have added additional text regarding secondary effects of CCCP to the Results section (lines 184-187).

3rd Editorial Decision

14 July 2017

I am very pleased to accept your manuscript for publication in the next available issue of EMBO reports. Thank you for your contribution to our journal.

YOU MUST COMPLETE ALL CELLS WITH A PINK BACKGROUND

Corresponding Author Name: Vicki Gold
 Journal Submitted to: EMBO Reports
 Manuscript Number: EMBOR-2017-44261V1